# Long-range charge transfer mechanism of the III₂IV₂ mycobacterial supercomplex

Daniel Riepl [1,2], Ana P. Gamiz-Hernandez [1,2], Terezia Kovalova[1,2], Sylwia M. Król[1], Sophie L. Mader [1], Dan Sjöstrand[1], Martin Högbom [1], Peter Brzezinski [1] & Ville R. I. Kaila [1] ✉

Aerobic life is powered by membrane-bound redox enzymes that shuttle electrons to oxygen and transfer protons across a biological membrane. Structural studies suggest that these energy-transducing enzymes operate as higher-order supercomplexes, but their functional role remains poorly understood and highly debated. Here we resolve the functional dynamics of the 0.7 MDa III₂IV₂ obligate supercomplex from *Mycobacterium smegmatis*, a close relative of *M. tuberculosis*, the causative agent of tuberculosis. By combining computational, biochemical, and high-resolution (2.3 Å) cryo-electron microscopy experiments, we show how the mycobacterial supercomplex catalyses long-range charge transport from its menaquinol oxidation site to the binuclear active site for oxygen reduction. Our data reveal proton and electron pathways responsible for the charge transfer reactions, mechanistic principles of the quinone catalysis, and how unique molecular adaptations, water molecules, and lipid interactions enable the proton-coupled electron transfer (PCET) reactions. Our combined findings provide a mechanistic blueprint of mycobacterial supercomplexes and a basis for developing drugs against pathogenic bacteria.

Aerobic respiratory chains comprise membrane-bound redox enzymes that transfer electrons to oxygen and pump protons across a biological membrane[1], powering the synthesis of ATP[2]. However, unlike the mammalian electron transport chains (ETCs), which shuttle electrons from Complex I to Complex IV, bacteria employ highly branched ETCs that vary in their composition and operational mode depending on the external conditions[1].

The enzymes responsible for the ETC can operate as higher-order supercomplexes (SCs)[3–5], although their functional role remains elusive[4,6–9]. The architecture of several SCs with different stoichiometries of Complexes I, II, III, and IV, were recently resolved[10–21], including the mycobacterial III₂IV₂ SC from *M. smegmatis*[10–14] and related species[15,16,22] that forms an obligate assembly operating only as a single functional unit. Mycobacteria comprise several pathogens that cause serious diseases such as tuberculosis, with major impact on global health[23]. Understanding

the bioenergetic principles of this mycobacterial SC could thus provide new avenues for developing drugs against emerging multi-resistant bacteria.

Recent cryo-electron microscopy (cryo-EM) studies of the III₂IV₂ SC from *M. smegmatis* revealed a 0.7 MDa dimeric protein complex, comprising up to 26 subunits and stabilised by cardiolipin at the dimer interface[10,12,13] (Supplementary Fig. 1a). The structures also resolved partial density for menaquinone molecules in non-canonical binding sites (here Q₀₂ and Q₀₁ᵦ), and inhibitors in the Q₀₁ₐ site, including Q203 and TB47[12,13], currently used as drugs against tuberculosis.

The mycobacterial SCs harbour several adaptations that could be of functional relevance. *M. smegmatis* lacks genes for a soluble cytochrome *c* and employs instead the QcrC subunit to shuttle electrons between Complexes III and IV (Fig. 1), thus making the SC assembly necessary for its proper function (Supplementary Fig. 1h). The SC also features a C-type superoxide dismutase (SodC) and subunit LpqE, both

---

[1]Department of Biochemistry and Biophysics, The Arrhenius Laboratories for Natural Sciences, Stockholm University, SE-106 91 Stockholm, Sweden. [2]These authors contributed equally: Daniel Riepl, Ana P. Gamiz-Hernandez, Terezia Kovalova. ✉e-mail: ville.kaila@dbb.su.se

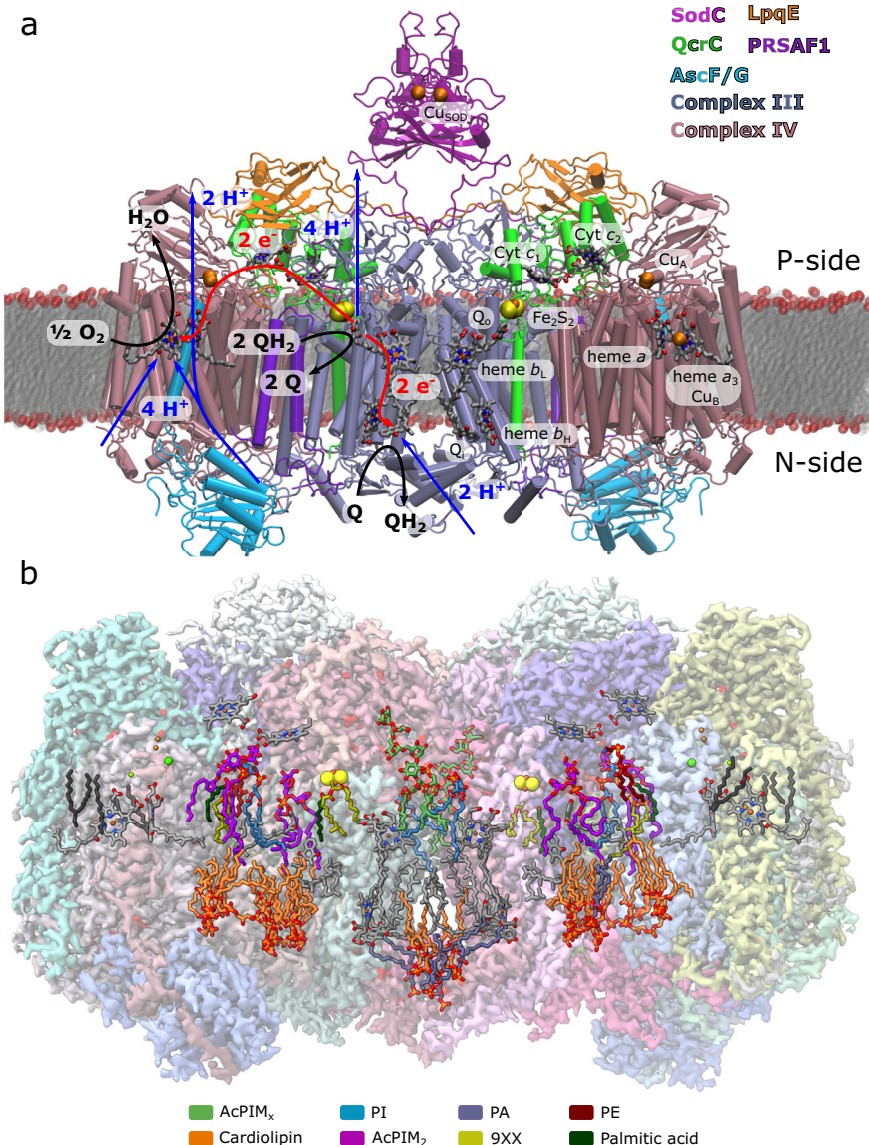

**Fig. 1 | Structure and function of the mycobacterial supercomplex III₂IV₂ from M. smegmatis. a** The figure shows all subunits, with the Complex III dimer depicted in iceblue and Complex IV in light pink, QcrC (cyt $c_1c_2$) in green, and PSRFA1 (supercomplex assembly factor) in violet. The AscF/G subunit is shown in cyan, and LpqE in orange, and the superoxide dismutase (SodC) in magenta. LpqE and SodC are anchored to the membrane by a lipid-modified cysteine. The SC catalyses quinol (QH₂) oxidation in Complex III coupled to oxygen reduction in Complex IV in a process linked to proton translocation (see main text). The figure shows putative electron and proton transfer pathways and stoichiometries of one half-reaction within one protomer of the SC. **b** Overview of the resolved 2.3 Å cryo-EM structure of *M. smegmatis* SC, with a local resolution of 2.0 Å for the core regions (see Supplementary Fig. 15). The density is shown at a threshold of 0.25. All cofactors are coloured in grey, whereas resolved lipids are coloured by the lipid type. The cryo-EM structure shows multiple resolved acylated phosphatidylinositol mannoside (AcPIMx) lipids that form the main component of the mycobacterial inner membrane. AcPIMx ($x > 4$) lipids are shown in green, phosphatidylinositols in blue, phosphatidic acid in iceblue, phosphatidylethanolamine in chestnut, cardiolipin in orange, AcPIM₂ in purple, (2S)-1-(hexadecanoyloxy)propan-2-yl (10S)-10-methyl-loctadecanoate (9XX) in beige, and palmitic acid in dark green.

of which are anchored to the membrane by lipid modifications (Supplementary Fig. 1c, d). The SC was observed in both symmetric[11,13] and asymmetric assemblies[10,12] with respect to the QcrC module, establishing contacts with either both or one of the Complex III and IV domains, respectively. The symmetric model features LpqE, as well as two closed QcrCs that bridge between the complexes, while the asymmetric model lacks LpqE and results in an open QcrC with a disconnected electron transfer pathway (Supplementary Fig. 1a, b)[10,11]. While dissociation of LpqE is likely to arise from purification artefacts, it cannot be fully excluded that switching into an asymmetric state could also provide a possible regulatory function.

Despite the structural differences that are likely to tune the energetics of the charge transfer process (see below), the conserved structural elements[5] of the mycobacterial SC, suggest that it could utilise overall similar charge transfer pathways as Complex III and IV[1,24–26] variants, not found in obligate bacterial SCs (from here on referred to as canonical Complexes III and IV, Fig. 1). In this regard, the Complex III module of the SC could employ the well-established Q-cycle mechanism[27,28], in which the Qₒ site bifurcates the electrons from menaquinol (instead of ubiquinol used in the canonical Complex III) to the Rieske FeS centre and further to the QcrC domain, whereas the other electron transfer branch shuttles the electrons via the low and high potential hemes $b_L$ and $b_H$ to the Qᵢ site, where another quinone is stepwise reduced to quinol (Fig. 1). The quinol oxidation at Qₒ releases the protons to the positively-charged (P-side) of the membrane, whilst quinone reduction at the Qᵢ site results in proton uptake from the

N-side. However, due to the unique mycobacterial subunits, the proton pathways likely follow different routes as those revealed in prior structures of the canonical isoforms[29,30]. The complete turnover of the Complex III module thus involves two quinol oxidation steps at the $Q_o$ site and one quinone reduction at the $Q_i$ site that establishes a proton gradient across the membrane.

The electrons released at the $Q_o$ site are transferred via the QcrC domain (cyt $c_1c_2$) to $Cu_A$ of Complex IV, without a soluble cyt $c$[10–14] (Supplementary Fig. 1h), which also implies that the mycobacterial SC catalyses electron bifurcation without the characteristic motion of the Rieske domain[31,32]. In this regard, experiments[33–35] and recent DFT studies[36] revealed insight in the proton-coupled electron transfer (PCET) steps between the quinol and the FeS Rieske centre for the canonical Complex III, while the PCET mechanism in the mycobacterial SC remains poorly understood. The electrons then continue from the QcrC domain and $Cu_A$, further to heme $a$ and the binuclear heme $a_3$/$Cu_B$ centre (BNC), responsible for oxygen reduction to water[24]. Prior data on Complex IV[24,37] show that the inter-heme electron transfer couples to proton uptake via the D- and K-channels (named after the conserved Asp115 and Lys340 in *M. smegmatis* numbering) that also have modular adaptations as compared to the canonical enzyme. These channels shuttle the protons both to the active site and across the membrane[24]. The Complex IV domain of the SC is expected to operate as a redox-driven proton pump, in contrast to the Complex III domain that generates *pmf* via the redox-loop mechanism (cf. refs. 1,26 for discussion on different proton translocation mechanisms).

To address the functional dynamics responsible for this fascinating long-range charge transfer process in the mycobacterial III₂IV₂ SC, we integrate here structural, functional, and computational methods. More specifically, we aim to address principles of quinone binding in the SC, the mechanism and energetics of the electron bifurcation process, how the formation of functional water wires enables the catalysis and proton release, and how modular adaptions in the catalytic site could tune the overall energetics. To this end, we combined large-scale atomistic molecular dynamics (MD) simulations and hybrid quantum/classical (QM/MM) free energy simulations with high-resolution cryo-EM analysis and biochemical activity assays, allowing us to study the link between catalysis and conformational dynamics. Our high-resolution structures (2.3 Å, 2.8 Å) together with multi-scale simulations and functional assays resolve unique proton pathways, quinone binding sites, and position of lipid molecules of the mycobacterial SC. These findings establish structural and mechanistic understanding of the PCET energetics and redox tuning effects, and they explain how the mycobacterial Complex III catalyses electron bifurcation. Importantly, the differences to the well-studied canonical complexes (cf. refs. 1,24–26,32) provide an important basis for drug design against pathogenic bacteria.

## Results
### Global dynamics and proton pathways of the SC
To probe the functional dynamics of the *M. smegmatis* III₂IV₂ SC, we constructed a symmetric model with respect to the QcrC subunit based on cryo-EM data (see Methods). The SC model was embedded in a lipid membrane, and modelled in different redox and protonation states, with the Q species modelled in the canonical ($Q_{o1a}$) and non-canonical sites around $Q_o$ ($Q_{o1b}$, $Q_{o2}$) (Fig. 2). Moreover we modelled a menaquinone in the $Q_i$ site, and the cofactors of Complex IV in their oxidised states[24] (see Methods, Supplementary Figs. 1 and 2, Tables 1–3, and Methods for detailed simulation models, redox and protonation states). The total system comprised around 1 million atoms, which we explored by ca. 10 µs atomistic MD simulations in different states along the reaction cycle (Supplementary Table 1, Movie 1). While the focus here is on the symmetric SC, we also conducted MD simulations on an asymmetric SC model (see above, Methods) to probe how the possible open QrcC conformation could

affect interactions with SodC and the electron transfer dynamics. To experimentally validate our findings, we performed cryo-EM experiments, where we refined a glyco-diosgenin (GDN) detergent-solubilised, symmetric SC to an overall resolution of 2.3 Å, and a local resolution of 2.0 Å for the core parts, and another dataset of the same state, to 2.8 Å global resolution (to 2.5 Å at the core), enabling us to independently determine the position of functionally central and previously unresolved water molecules that are key for the proton transfer reactions. We focus here on the higher resolution dataset, which contains more details, while the other dataset shows overall similar results and serves as an independent validation of our model. We also characterised the quinol oxidation-$O_2$ reduction activity by biochemical assays that we compared with kinetic network models of the long-range charge transfer process.

The global dynamics extracted from the MD simulations of the SC closely resemble the variation in the local resolution in both our current and prior cryo-EM maps (Supplementary Fig. 3). The lipid-anchored SodC module forms the most dynamic part of the SC in the MD simulations, consistent with the blurred density observed in our cryo-EM data (Supplementary Fig. 4). The SodC samples distances in the range of ca. 40–80 Å from the QcrC domain in the symmetric model, and ca. 25–50 Å for the asymmetric model (Supplementary Figs. 1, 4b). Other highly dynamic regions include QcrC (open conformation), AscF/G (previously MSMEG 4692/4693[22]), and LpqE, whereas the dimeric SC interface, formed by cardiolipin molecules (see below), remains highly stable during the simulations (Supplementary Figs. 1 and 3).

The SC becomes highly hydrated in functionally important regions during the MD simulations, leading to a total influx of around 900 water molecules in the buried parts of the protein (Supplementary Fig. 5) that could not be resolved in prior cryo-EM studies of the *M. smegmatis* SC (but cf. ref. 15). However, our current high-resolution cryo-EM analysis revealed around 600 functionally relevant water molecules (at 2.3 Å, ca. 200 for the 2.8 Å map) that compare well with the data from our MD simulations (see below). More specifically, our cryo-EM data show 20–30 water molecules between the $Q_{o1a}$ site and QcrC that are in excellent agreement with our MD simulations (Fig. 3c, Supplementary Figs. 5 and 6). These water molecules establish a hydrogen-bonded pathway from the $Q_{o1a}$ site via Tyr159, Asp302, and His110 to the P-side, whereas the other branch connects to the P-side via His355, Asp309, and Arg313 (Fig. 3a, b, Supplementary Fig. 6a). From the N-side of the membrane, we also observe a putative proton pathway from Lys260 and Glu44 to $Q_i$ in both the MD simulations and the cryo-EM data (Fig. 3d, i).

On the Complex IV side of the SC, our combined MD and cryo-EM data show hydrogen-bonded water arrays along the D- and K-channels that support the proton uptake from the N-side of the membrane to Glu266 (via the D-channel) or via Lys340 to Tyr268 of the BNC (via the K-channel) (Fig. 4d–f)[38,39] (cf. also refs. 15,40,41). Glu266, the terminal residue of the D-channel samples both downward and upward conformational states (Fig. 4b) in the MD simulations that favour contacts with the respective D-channel and a putative proton-loading site (PLS) near the propionates of heme $a_3$, that transiently stores protons during proton pumping across the membrane in canonical Complex IV (cf. ref. 42). When Glu266 is in the upward conformation, the non-polar cavity next to the BNC occupies around 6 water molecules during the MD simulations, with Glu266 forming proton wires both to the propionate groups of heme $a_3$ or to the BNC (Fig. 4a)[24,40–43] that could direct the protons for pumping or to be consumed in the reduction of $O_2$ to $H_2O$, respectively[40,41]. In contrast, the downward conformation is less hydrated in the MD simulations and lacks clear proton pathways between Glu266 and the BNC. Our cryo-EM data show a well-resolved density of the entire sidechain of Glu266 in the downward conformation (Supplementary Fig. 7c), possibly due to the protonated form of the residue, and further supporting that the residue has a high p$K_a$

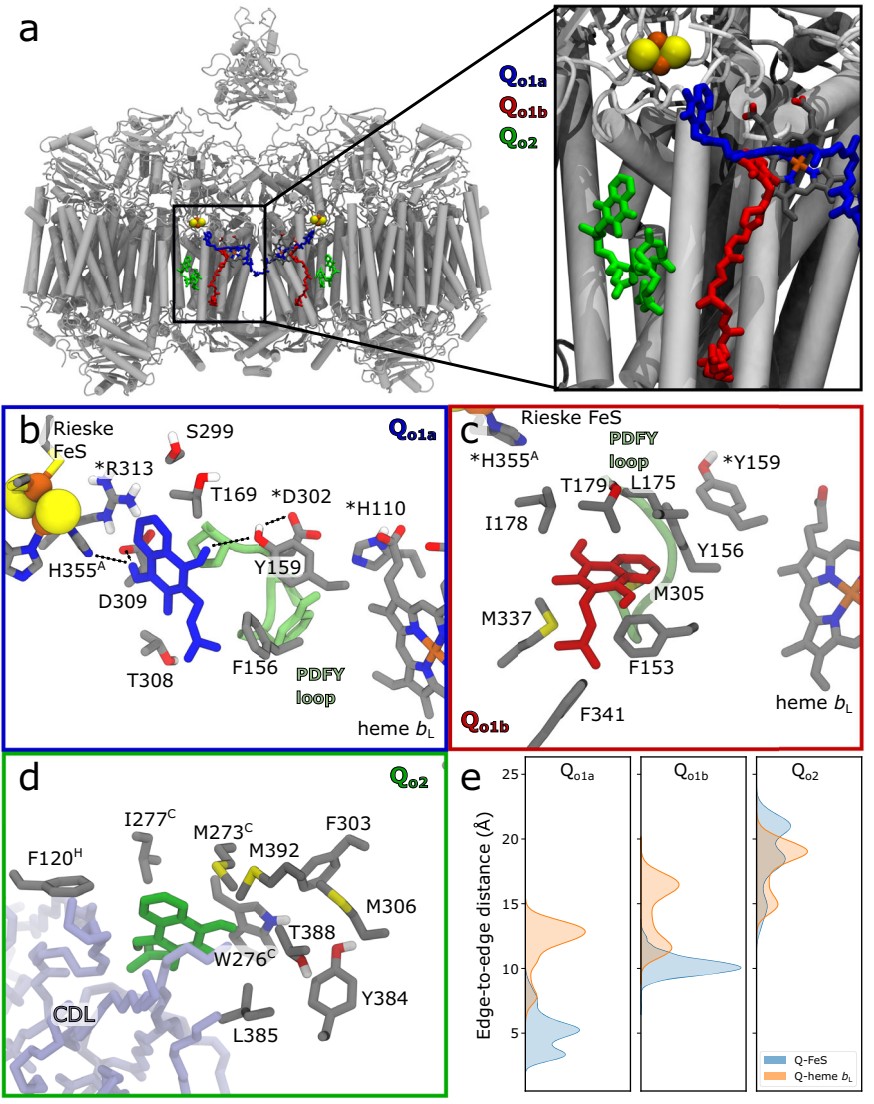

**Fig. 2 | Quinone binding sites at the P-side of the membrane in the SC.**
**a** Overview of the $Q_o$ site, modelled based on PDB ID: 6hwh and 6adq, and supported by our current cryo-EM models. The sites were modelled with menaquinols at the $Q_{o1b}$ and $Q_{o2}$ sites ((**c**), (**d**) and coloured in red, and green, respectively), based on resolved cryo-EM density, and at the canonical $Q_{o1a}$ site ((**b**), coloured in blue). The PDFY loop is shown in light green and CDL in light blue. Dashed lines highlight interactions that could be functionally relevant for catalysis. Superscripts denote residues from subunits other than QcrB, whilst an asterisk indicates residues, which do not interact directly with Q during the simulations. Refer to Supplementary Fig. 8 for distance between the Q and nearby residues during the MD sinulations. **e** Edge-to-edge distances between the Q sites/FeS Rieske centre and between Q / heme $b_L$ based on MD simulations. The simulations suggest that the menaquinol is dynamically more stable at the $Q_{o1a}$ site, as compared to the $Q_{o1b}$ and $Q_{o2}$ sites.

value, consistent with its function as a proton shuttle (cf. refs. [42],[44]). We further note that while Glu266 samples both conformations in the MD simulations, the downward conformation is energetically preferred, consistent with the lack of cryo-EM density for the upward conformation. From the propionate region of heme $a_3$, we observe pathways that support proton release from the $Mg^{2+}$ site to the bulk P-side of the membrane, also in line with prior data on the proton release pathways in canonical Complex IVs[43],[45] (Fig. 4c, i, see section: Analysis of the high-resolution cryo-EM structure).

**Quinone binding modes enabling catalysis**
To obtain insight into the long-range electron transfer process, we next studied binding of menaquinol around the $Q_o$ sites (Fig. 2a–d). The $Q_{o1a}$ site corresponds to the canonical $Q_o$ site in cytochrome $bc_1$ with previously resolved inhibitors at this location[13],[15],[46] (cf. also ref. [32], see Methods for details of modelling menaquinol binding to this site). In our cryo-EM data, we observe densities for menaquinone only in the

$Q_{o1b}$ and $Q_{o2}$ sites, consistent with previous studies[10],[11],[13],[15],[22], although their functional roles for catalysis have remained unclear.

Our MD simulations suggest that menaquinol is stabilised in the $Q_{o1a}$ site by His355 and Asp309, and by water-mediated contacts with Tyr159, which in turn interacts with Asp302 of the characteristic PDFY motif[47],[48] (Figs. 2b, 3a–c, Supplementary Fig. 9). Asp302 establishes an ion-pair with His110, the conformation of which modulates the binding affinity of the substrate (Supplementary Fig. 9e). The water wires leading from the menaquinol to His110/Asp302 and Asp309/Arg313 are connected to the P-side bulk, forming possible proton exit pathways that are also well-resolved in our cryo-EM data, with water positions closely matching those observed in the MD simulations (Fig. 3a–c, Supplementary Figs. 6, 10).

In the $Q_{o1b}$ site, the menaquinol is at ca. 10 Å edge-to-edge distance from the FeS Rieske centre and somewhat further (12–17 Å) from heme $b_L$ (Fig. 2e). The menaquinol headgroup is stabilised by π-stacking interactions with Phe153, and non-polar contacts with, e.g.

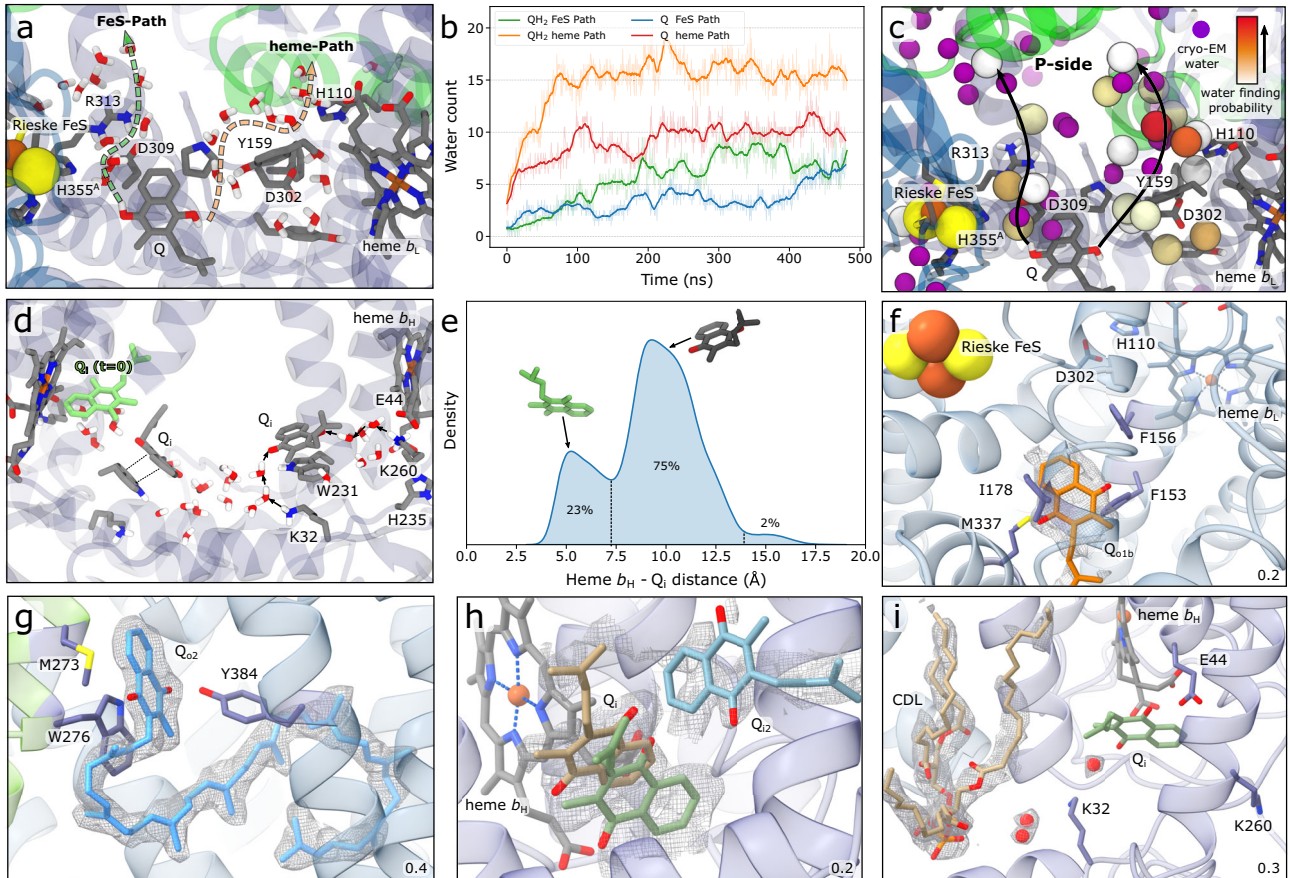

**Fig. 3 | Functional hydration dynamics of the Complex III module of the SC.**
**a** Menaquinol in the $Q_{o1a}$ site as predicted by the MD simulations and the putative proton pathways towards Asp309 (path 1, green) and Asp302 (path 2, orange) leading to the P-side bulk. **b** Water occupancies from MD simulations for $QH_2$ and $Q_{ox}$ (simulations S1/S7, Supplementary Table 1) along the pathways shown in panel a. Shaded lines show the raw data and solid lines the smoothed data. **c** MD-based water clustering (white to red colour scale) and resolved cryo-EM water molecules (purple) predict similar proton pathways from the $Q_{o1a}$ site to the P-side bulk. **d** The quinone at the $Q_i$ site forms an π-stacking interaction with Trp231 and two possible proton pathways from the N-side bulk via Lys260 and Lys32. The experimentally resolved starting position of the quinone is shown in light green. **e** The MD simulations predict two positions (distal in atom colours, proximal in green) for $Q_i$,

located around 5 Å and 10 Å from heme $b_H$, respectively. The two quinone positions shown in (**d**) are indicated with arrows. **f–i** Cryo-EM map and model of resolved cofactors and proton arrays, with the density thresholds shown in the lower right corner. **f** Menaquinone (orange) bound at the $Q_{o1b}$ site. **g** $Q_{o2}$ site located next to the QcrC helix where menaquinone (cyan) forms an π-stacking interaction with Trp276. **h** The $Q_i$ site density supports binding of menaquinone in three different conformations, two of which are adjacent to heme $b_H$ (beige, green), and one position somewhat (14 Å) further away ($Q_{i2}$, cyan). **i** The $Q_i$ site with a nearby cardiolipin (beige) and putative proton donors are shown. Water molecules connect the menaquinone to Lys32 and further to the cardiolipin. Only the menaquinone closest to heme $b_H$ is shown for visual clarity.

Met305 and Met337 (Fig. 2c, Supplementary Fig. 8a, b). In the $Q_{o1a}$ site the menaquinol is closer to the redox centres, ca. 5 Å edge-to-edge distance from the FeS Rieske centre, and around 12 Å from heme $b_L$ (Fig. 2b, e). Interestingly, in some simulations initiated from the $Q_{o1a}$ site, we observe a transient motion of the quinol towards the $Q_{o1b}$ site, suggesting that these regions are dynamically exchangeable, with comparable binding energies (Supplementary Fig. 8h–j).

Our MD simulations suggest that the menaquinol is dynamically flexible at the $Q_{o2}$ site, sampling several binding poses with a broad edge-to-edge distance distribution at ca. 20 Å between the quinone and heme $b_L$/FeS Rieske centre (Fig. 2e). The menaquinol forms π-stacking / hydrogen-bonding interactions with Trp276, Thr388, Phe303, and Tyr384 (Fig. 2d), consistent with the well-resolved density in the cryo-EM maps (Fig. 3g). Despite the stable binding mode (Supplementary Fig. 8h–j), we do not observe residues that could support the proton-coupled oxidation of the menaquinol, suggesting that $Q_{o2}$ is not directly employed for catalysis. Moreover, in contrast to the exchangeable $Q_{o1a/b}$ sites, we could not observe substrate tunnels connecting $Q_{o2}$ with the $Q_{o1a/b}$ sites, despite similar binding energies amongst the sites in the MD simulations (Supplementary Fig. 8h–j).

## Proton-coupled electron transfer dynamics in the $Q_o$ site

We next addressed the mechanism of the PCET reactions during the menaquinol oxidation by QM/MM free energy simulations, which allowed us to study the energetics of bond-breaking and formation at a quantum mechanical level (Methods, Supplementary Fig. 11). These simulations show that the proton transfer from the menaquinol to His355 in the $Q_{o1a}$ site takes place by a concerted PCET process, where the proton moves along the menaquinol-His355 hydrogen-bond and the electron is transferred from the aromatic ring of the quinol directly to the (His355-ligated) ferric iron of the FeS Rieske centre (Fig. 5a, e, Supplementary Fig. 11, Movie 2). The PCET reaction has a free energy barrier of around 5 kcal mol$^{-1}$ (Fig. 5c) and the process is weakly exergonic, consistent with redox mid-point potentials of the menaquinol[49] ($E_{m,7}^{exp} = -80$ mV) and the FeS centre ($E_{m,7}^{exp} = +160$ mV, $\Delta G^{exp} = -5.5$ kcal mol$^{-1}$; $\Delta G_{QM/MM} = -4.6$ kcal mol$^{-1}$), (cf. also refs. 36,50, Supplementary Table 4). After the initial H$^+$/e$^-$ transfer step, the resulting semiquinone (QH·) deprotonates via a water-bridge contact via Tyr159 to Asp302 (Fig. 5b, d, Supplementary Fig. 11, Movie 3). The second PCET process leads to a Grotthuss-type proton transfer reaction, with a barrier of +11 kcal mol$^{-1}$ (rate of 14 μs$^{-1}$) that triggers an

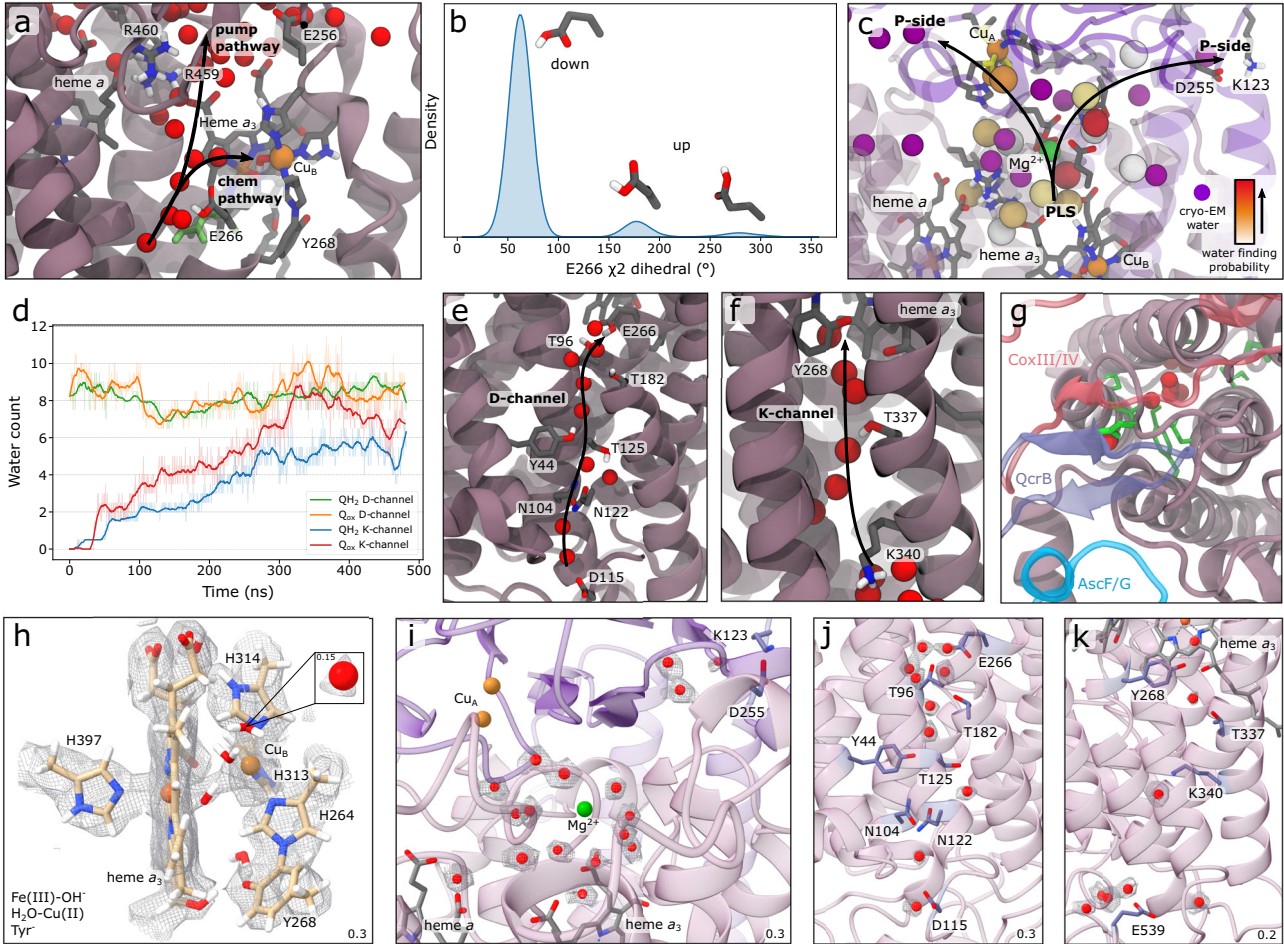

**Fig. 4 | Functional hydration dynamics of the Complex IV module of the SC.**
**a** Proton pathways connect Glu266 with the propionate site of heme $a_3$ and Glu266 with the BNC of the Complex IV domain. Glu266 is shown both in an upward (coloured by atom types) and downward conformation (green). **b** Glu266 samples in the MD simulations mainly the downward position, but with frequent transient flips into the upward position (Glu266 shown in same orientation as in (**a**). **c** Water clusters from simulations (white to red colour scale) and resolved cryo-EM waters (purple) predict putative proton pathways from the PLS to the P-side bulk. **d** Water count (raw data shown as shaded lines and smooth data as solid lines) and **e**, **f** hydration of the K- and D-channel of Complex IV during the MD simulations (simulations S1/S7, Supplementary Table 1). **g** The entrance of the D-channel in the

SC is partially disrupted by QcrB (blue), CoxIII/IV (red), and AscF/G (cyan). Channel residues are shown in green. **h**–**k** Cryo-EM map and resolved cofactors and water molecules, with density thresholds shown in the lower right corner (see *Methods*). **h** The BNC contains a cylindrical density connecting the iron of heme $a_3$ with $Cu_B$. The figure shows a quantum chemically optimised model of the $Fe^{III}$–$OH^-$/$H_2O$–$Cu^{II}$/ TyrO$^-$ state, overlayed on the cryo-EM density. See Supplementary Fig. 7 for alternative models. **i**, Structurally resolved water molecules show proton pathways from the PLS region towards the P-side bulk. **j** Resolved water molecules in the D-channel, and (**k**) in the K-channel, with distinct similarities to the hydration patterns predicted by our MD simulations (**e**, **f**).

electron transfer to heme $b_L$ (Fig. 5d, Supplementary Fig. 11). We note that the $Q_{o1b}$ site is unlikely to support a similar reaction due to the long distance between the Q and His355/Tyr159/Asp30 (see also Supplementary Table 4, Fig. 12).

As usual in biological electron bifurcations[51–54], this endergonic charge transfer step could be driven by the initial exergonic PCET step (Fig. 5d). The energetics suggests that the second PCET step is rate-limited by the proton transfer followed by a rapid non-adiabatic electron transfer reaction. The proton is directed from this PCET process to the His110/Asp302 ion-pair, which could hold the quinol proton until the electron has been transferred to the heme $b_H$ / $Q_i$ site. In this regard, electrostatics calculations (Fig. 6f) suggest that the oxidation of heme $b_L$ lowers the proton affinity of the His110/ Asp302 cluster, and favours the proton release to the P-side bulk via the water arrays described above (Fig. 3a, c, Supplementary Fig. 6a). These local PCET reactions further couple to the energetics of quinol binding and quinone release, whilst the overall process is driven by the exergonic oxygen reduction reaction at the BNC (see Supplementary Fig. 24 for schematic free energy profile). Taken together,

these findings suggest that the PCET-driven quinol oxidation is catalysed by the $Q_{o1a}$ site.

## Long-range electron transfer kinetics

To probe the long-range charge transfer pathways within the SC, we next developed a kinetic network model based on our multi-scale simulations, experimental redox potentials, reorganisation energies[26,49], as well as kinetic measurements[55,56] (see Methods, Supplementary Fig. 12). While the models explicitly account for electron transfer reactions, the quinol/quinone binding/unbinding (in Complex III) and water formation (in Complex IV) are implicitly accounted for by modelling the associated steps as irreversible. Our model suggests that the symmetric SC is in complete charge transfer contact from the $Q_{o1a}$ site of Complex III to the BNC of Complex IV, with an overall PCET rate within the SC of ca. 88 s$^{-1}$ (Fig. 6d), which is roughly 40 times slower relative to the canonical $aa_3$-type Complex IV[57,58], but compares very well with our experimental SC activity of around 90 s$^{-1}$ of GDN-solubilized III$_2$IV$_2$ SC (Fig. 6c, e) (cf. also ref. 13). In this regard, our model accounts for the proton-coupled reduction of the BNC[41] and the

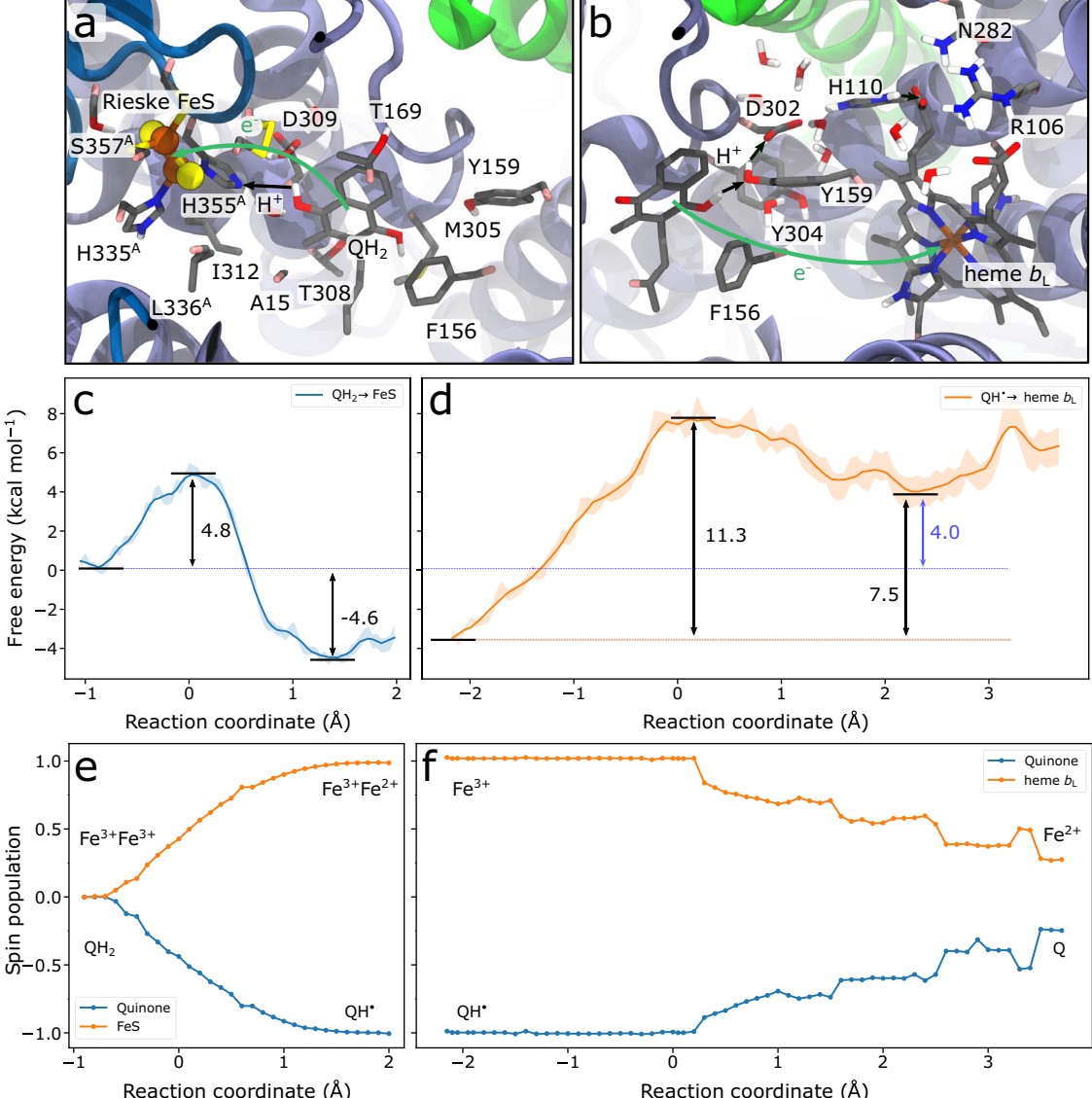

**Fig. 5 | PCET energetics in the $Q_{o1a}$ site.** QM regions used for the hybrid QM/MM free energy simulations, sampling (**a**) the first PCET ($QH_2 \rightarrow QH^{\bullet}$) and (**b**) the second PCET ($QH^{\bullet} \rightarrow Q$) reactions. Black arrows depict the proton transfer reaction coordinate (see also Supplementary Fig. 11). **c** QM/MM free energy profiles for the initial exergonic PCET (*left*, blue) leads to a proton transfer from menaquinol to the His355 ligand of the Rieske FeS centre, coupled with a concerted electron transfer to the nearest Fe of the FeS Rieske centre. **d** The second, endergonic step (*right*,

orange) comprises an electron transfer from mena-semiquinol ($QH^{\bullet}$) to Asp302 via Tyr159, coupled an electron transfer to heme $b_L$, which results in the formation of the oxidised menaquinone. Shaded areas denote the statistical uncertainty based on Monte Carlo bootstrapping. Spin populations for (**e**) the first PCET reaction show electron transfer from menaquinol to Rieske FeS centre forming mena-semiquinol, followed by electron transfer from (**f**) $QH^{\bullet}$ to heme $b_L$.

experimentally measured proton uptake rate in the SC Complex IV of ~4 ms[55,56] that tune the apparent redox potential of the BNC (see Supplementary Information). Without such redox gates, the model predicts a pure electron transfer flux in the SC of 1000–2000 s⁻¹, i.e. an order of magnitude faster rate as compared to our experimental observations (Fig. 6a, Supplementary Fig. 13, Table 4). Interestingly, however, our kinetic model suggests that the electron initially reaches a pre-steady state, with the electron distributing on heme *a*, $Cu_A$, and heme $c_2$ centres until the proton uptake from the D-channel tunes the redox midpoint potential of heme $a_3$/$Cu_B$ that completes the reaction (Fig. 6b). The complete turnover of the canonical Complex III takes place with a rate of ca. 120–300 s⁻¹[59–61], whilst our kinetic models and activity data suggests that mycobacterial Complex III is not rate-limiting for the long-range charge transfer process, although it has not been possible to experimentally determine uncoupled activities and potential rate-limiting steps in the mycobacterial Complex III. Overall,

these findings suggest that the proton uptake is rate-limiting for the overall long-range PCET network within the SC, with the observed turnover determined by the proton uptake to Complex IV.

To further address the electron transfer within the SC dimer, we analysed our MD simulations of the asymmetric SC, where the QcrC interface remains in an open conformation (Fig. 1a, Supplementary Fig. 1), consistent with previous structural data[10,12]. This, in turn, prevents electron transfer on physiologically relevant timescales on one side of the dimer, whilst we observe a similar electron transfer flux as in the symmetric model in the closed protomer (Supplementary Fig. 13d). Our models also predict a slow electron transfer rate relative to the overall turnover[62] (but cf. also ref. 34) between the Complex III protomers via the heme $b_L$–heme $b_L$ bridge (ca. 10 s⁻¹, Supplementary Fig. 13e, f).

Purification of the SC with *n*-dodecyl-β-D-maltoside (DDM) leads to dissociation of LpqE from the SC, which is supported by structural

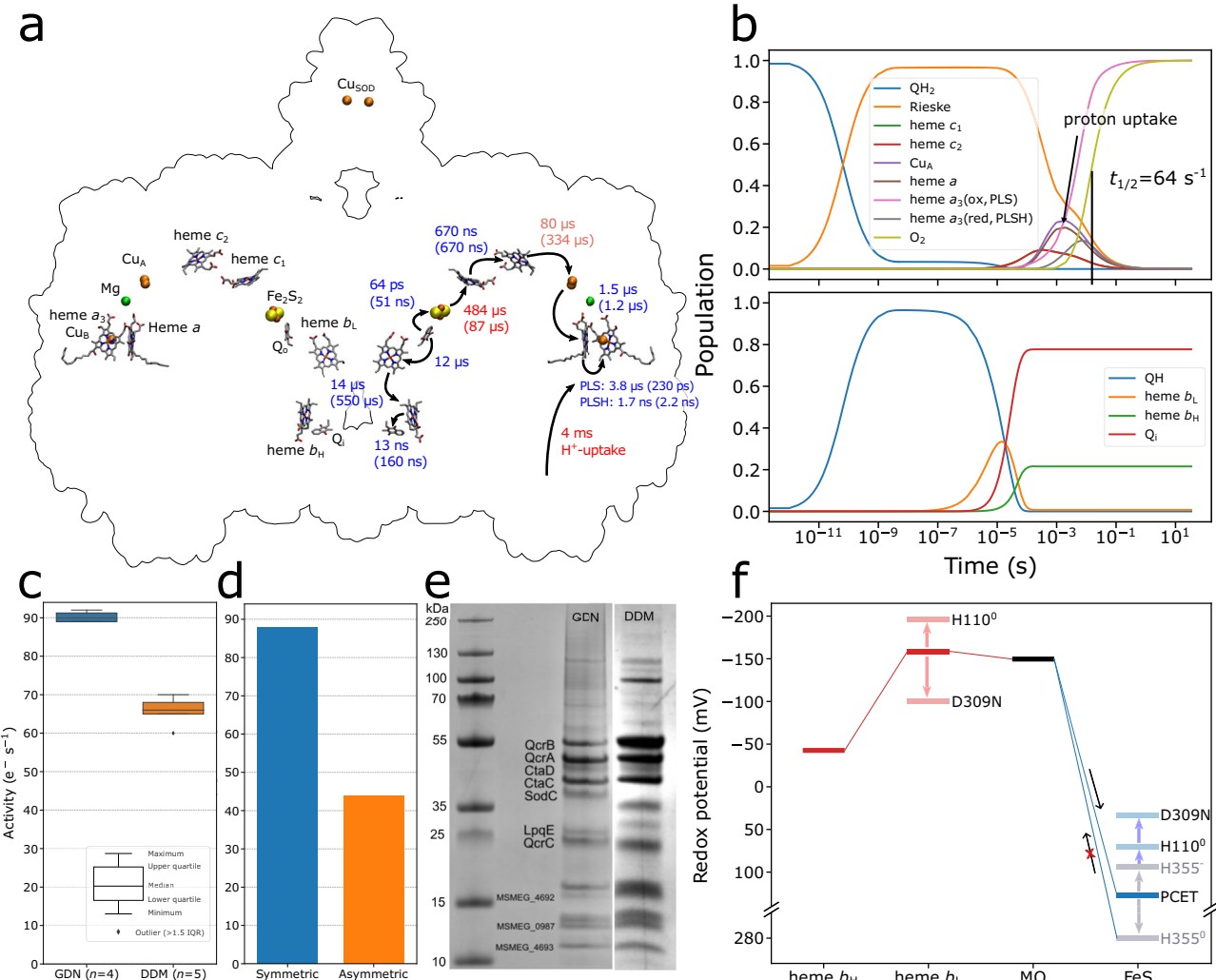

**Fig. 6 | Kinetic network and mechanistic model of the long-range charge transfer reactions in the III₂IV₂ SC from *M. smegmatis*. a** Average electron transfer rates (back transfer rates in parenthesis) calculated from simulation S1 (see *Methods* and Supplementary Information) with bottleneck rates for the electron transfer between the Rieske FeS centre and heme $c_1$. **b** Kinetic model of the closed SC protomer with a modelled 4 ms proton uptake in Complex IV that becomes the rate-limiting step. **c** Activity of the GDN- and DDM-solubilized SC. The number of technical repeats of the measurements is given in parentheses. **d** Rates calculated from the kinetic model in (**b**) show good overall agreement with the measured activities (see also Supplementary Fig. 13). **e** SDS-PAGE of the SC shows that some LpqE could still be present in the DDM-solubilised preparation, whilst the cryo-EM

map reveals a weak density of LpqE, suggesting that only a subfraction of the ensemble could have bound LpqE, in contrast to the GDN-solubilized SC. The absence of bound LpqE in the SC is expected to support the open QrcC conformation (Supplementary Fig. 2), with a disconnected electron transfer pathway between Complexes III and IV. **f** Changes in calculated mean redox midpoint potentials (relative to the WT model) upon mutation of key residues around the $Q_{o1a}$ site. The D309N mutation downshifts the Rieske FeS-redox potential, while upshifting the potential of heme $b_L$, whereas protonation of His110 has a small downshifting effect on both cofactors (see Supplementary Fig. 19 and Supplementary Table 5 for further details).

data showing only a weak density for the subunit. Small amounts of the LpqE subunit remain in the sample or associated to the SC (Fig. 6e, see also ref. [12]), and could explain why the DDM preparation shows only around 30% reduction of the SC activity (66 s⁻¹) relative to the GDN-solubilized SC, where all subunits are present (Fig. 6c), instead of a 50% reduction as may be expected, if one electron transfer branch would be completely blocked. The reduced activity is likely to arise from a significant amount of open QcrC conformations, consistent with previous structural data[10] and our kinetic models (Fig. 6a, b).

On the acceptor side of Complex III, we observe a subtle motion of the $Q_i$ quinone from the resolved binding position to a site, ca. 4 Å further from heme $b_L$, where the quinone forms a π-stacking interaction with Trp231 in the MD simulations (Fig. 3d, e). The heme $b_L$–heme $b_H$–$Q_i$ network favours rapid electron transfer (10³–10⁴ s⁻¹, Fig. 6a), with our QM/MM simulations (Supplementary Fig. 14) further suggesting that Lys260/Glu44 or Lys32 could function as local proton

donors upon the quinone reduction (Supplementary Fig. 14, Movie 4), following uptake of the protons from the N-side via the nearby cardiolipin gate (see below, and cf. also refs. [30,63]). In this regard, our simulations suggest that proton transfer may only occur after the $Q_i$ species has been fully reduced (Supplementary Fig. 14, Movies 4, 5). Our cryo-EM data revealed a well-defined density of a cardiolipin molecule and two water molecules near the $Q_i$ site that support the positioning of the $Q_i$ headgroup and the putative cardiolipin lipid-mediated proton uptake route (Fig. 3i). These pathways are readily accessible from the N-side bulk, and therefore not expected to be rate-limiting for the overall long-range charge transfer process.

## Analysis of the high-resolution cryo-EM structures
The two cryo-EM datasets of the *M. smegmatis* III₂IV₂ SC allowed us to gain experimental insight into the SC function and validate our computational findings (Fig. 1b, Supplementary Figs. 15, 16). The refined

structures revealed a symmetric SC, comprising 22 main subunits, including LpqE. The data allowed us to assign densities of 614 water molecules (in the 2.3 Å map, and 249 in the 2.8 Å map, Supplementary Fig. 10), many of which are located around key functional regions in Complexes III and IV, as discussed above. The two datasets are consistent with each other, resolving water molecules at the same positions in several sites, but with significantly higher statistical certainty in the 2.3 Å resolution map. Importantly, the higher (2.3 Å) resolution map allowed us to further assign functionally central water molecules near $Cu_B$, as well as in the D- and K-channels. These experimentally determined water positions are in excellent overall agreement with the hydration dynamics independently predicted by our MD simulations, supporting the complementarity between the approaches, but not observed in the 2.8 Å map due to possible averaging effects.

The D-channel shows 10 water molecules connecting Asp115 with Glu266 (Fig. 4e, j), supporting the proton pathway in the D-channel. The non-polar cavity next to the BNC that establishes a proton wire between Glu266 and $Cu_B$ in the MD simulations (Fig. 4a), shows a weak cryo-EM density for a water molecule near $Cu_B$ (Fig. 4h, inset). The K-channel harbours five water molecules at the channel entrance close to Glu539, as well as water molecules next to Lys340 (Fig. 4k), between Thr337, Tyr268, and the BNC that could support the proton uptake. Overall, more water is present in the D-channel as compared to the K-channel. This could, at least in part, arise from the larger volume of the D-channel (Supplementary Fig. 18d), but also from the higher water mobility in the K-channel, as suggested by the MD simulations (Supplementary Fig. 18e). We note that prior high-resolution structures (PDB ID: 7COH[64], 7ATE[65], 7AU6[65]) of the canonical Complex IV support similar hydration levels. Interestingly, key residues along a previously proposed H-channel[66] are also present in the *M. smegmatis* SC, but both our cryo-EM data and MD simulations show low hydration of the region, suggesting that it is unlikely to conduct protons (cf. also ref. 67).

The heme $a_3$ iron of the BNC has a spherical density, and the shape of the porphyrin ring is well-resolved in the 2.3 Å map (Fig. 4h, Supplementary Fig. 7e–j). We could assign a Fe-Cu distance of 4.4 Å, and the continuous density connecting the metals could fit, e.g. a hydroxyl and a weak water ligand, based on QM/MM calculations, although assignment of other ligands is also possible (Fig. 4h, Supplementary Fig. 7e–j) (cf. also[65,68,69]). The characteristic His264-Tyr268 crosslink is well-resolved (Fig. 4h), with a refined C−N bond distance of 1.43 Å (based on QM/MM), and the heme $a_3$ farnesyl further stabilising the phenol group of Tyr268. The non-polar cavity is occupied by one water molecule with a rather low cryo-EM density (Fig. 4h), possibly due to the downward conformation of Glu266, which also leads to a low hydration level in the MD simulations. We further note that high-resolution x-ray and cryo-EM structures of the canonical Complex IV (PDB IDs: 7COH[64], 7ATE[65], 7AU6[65]) show similarly a dry cavity, together with Glu266 oriented towards the D-channel.

The propionates of heme $a_3$ comprise several water molecules that connect to the P-side bulk (Fig. 4i, c, Supplementary Fig. 6b, 10). We further resolved a $Ca^{2+}$ ion in pentagonal bipyramidal coordination around 12 Å (edge-to-*edge*) from heme $a$ (Supplementary Fig. 7b, d), in a position that could affect the redox properties of heme $a$, and/or the proton release along the exit pathway (cf. refs. 70–72).

On the Complex III side of the SC, we observe ten menaquinone molecules (five per monomer) at five different positions. The canonical $Q_{o1a}$ site is empty (cf. also refs. 10,12,13), but a menaquinone occupies the $Q_{o1b}$ site (Fig. 3f). Another menaquinone binds next to Trp276 in the $Q_{o2}$ site at the interface of Complexes III and IV (Fig. 3g), with an additional site close to the P-side (Fig. 1b), as also suggested in previous studies[10,12,13]. We also observe density corresponding to two menaquinones at the N-side of the membrane, one at the canonical $Q_i$ site, and one in another position, $Q_{i,2}$, ca. 10 Å away from the former (see Fig. 3h), with possible substrate queuing or regulatory functions.

The former site allows modelling of the quinone headgroup in alternate positions, with the weaker density suggesting that the $Q_{i,2}$ site has a lower affinity (cf. also PDB ID: 7E1V[13]). Due to the large distance from the residues supporting PCET during the $Q_i$ reduction, this accessory site is, however, unlikely to be involved in catalysis. We note that the quinones at the $Q_i$ site show partially overlapping densities, which could arise from an ensemble average of different binding modes.

Our resolved SC structures show multiple lipid-binding sites (Fig. 1b, Supplementary Fig. 17), including cardiolipin molecules on the N-side at the interface between the Complexes III and IV that could have a central structural role, as also suggested for other SCs[73]. We find further cardiolipins near the $Q_i$ site that are in different positions from those previously described for the mammalian Complex III[63], in place of which the *M. smegmatis* SC harbours the QcrC helix. One of these cardiolipins establishes a contact via water molecules and Lys32 to $Q_i$ and could catalyse the proton uptake from the N-side (Fig. 3i). The interface between Complexes III and IV contains four cardiolipin molecules on each side, three cardiolipins between PRSAF1 and Complex IV, and one cardiolipin next to subunit III of Complex IV (Fig. 1b). Two acyl phosphatidylinositol-hexamannoside (AcPIM$_6$) molecules, and four phosphatidylinositol (PI) lipid molecules are located near $Q_{o1a/b}$, the QcrB/PRSAF1 and the QcrA/QcrB interfaces (Fig. 1b, Supplementary Fig. 17), with the AcPIM$_2$ molecules next to the QcrC helix (Supplementary Fig. 7a), serving a possible structural role. The PIM lipids are characteristic for mycobacterial membranes[74] and likely to affect the biophysical properties of the bacterial membrane and/or the function of the SC. The Ac-PIMs are resolved on the P-side of the SC, consistent with previous assignments suggesting that mycobacterial membranes have an asymmetric lipid distribution with the P-side leaflet enriched by PIM$_6$ lipids[74].

## Discussion

Our integrative structure-function approach revealed key water structure, lipid interactions, and modular adaptations that govern the long-range PCET dynamics in the III$_2$IV$_2$ SC from *M. smegmatis*. Our combined findings show that the SC establishes an electronic wiring between the $Q_{o1a}$ site of Complex III and the active site of Complex IV, leading to a turnover rate of ~90 s$^{-1}$. Our large-scale MD simulations and high-resolution cryo-EM structures, resolved over 600 water molecules, which enable the proton release from the $Q_{o1a}$ site, proton uptake to the $Q_i$ site, as well as proton transfer across both the D- and K-channels to the oxygen reduction site and across the membrane. These proton transfer reactions support the redox-loop and redox-driven proton pumping mechanisms of the Complex III and IV modules, respectively.

On the Complex III side of the SC, the modular adaptations at the $Q_{o1a}$ site have important redox tuning effects that modulate the PCET reactions both to the FeS Rieske centre and to heme $b_L$ (Fig. 7). The $Q_{o1a}$ site catalytically supports these reactions, whereas that unique $Q_{o1b}$ site (Fig. 2e) may represent a transient substrate docking position, which can exchange with the $Q_{o1a}$ site. In contrast, the $Q_{o2}$ site is not structurally connected to $Q_{o1a}$, and the site lacks residues that could support the menaquinol oxidation. The $Q_{o1b}$ and $Q_{o2}$ sites are resolved in our high-resolution cryo-EM structures, and are consistent with previous data[10,12,13], whilst the properties of the $Q_{o1a}$ site were explored by multi-scale simulations. The water wires observed in our cryo-EM structures and MD simulations support the proton release upon the menaquinol oxidation via the QcrB/QcrC interface to the P-side of the membrane (Figs. 3a, c and 7) (cf. also ref. 15).

The mycobacterial $Q_o$ site could have evolved to account for the shifted redox properties of menaquinol relative to ubiquinone (Fig. 3f), as suggested by in silico mutagenesis and electrostatic models (see Supplementary Information). These calculations suggest that Asp309 has an important redox tuning effect that favours the PCET reaction: in the D309N variant, the redox potential of the FeS Rieske centre is

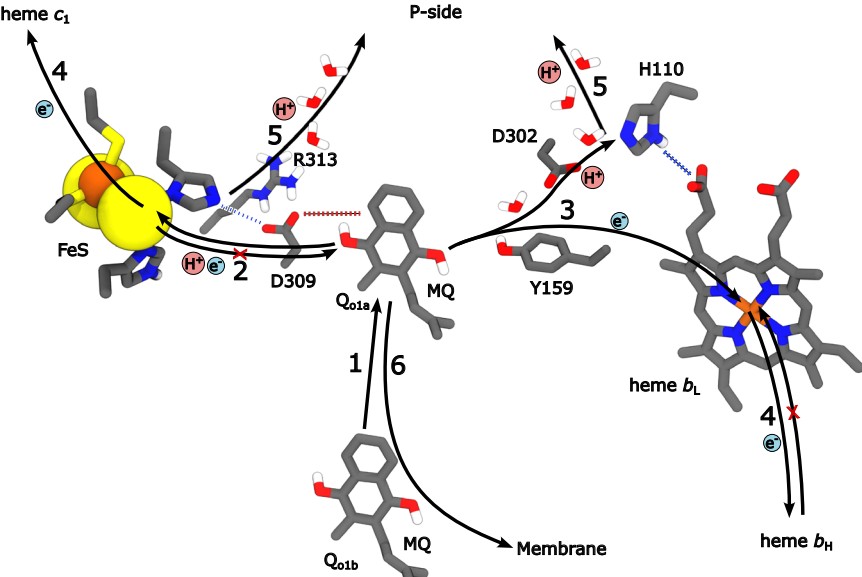

**Fig. 7 | Putative mechanism of quinol oxidation in the III$_2$IV$_2$ SC from *M. smegmatis*.** 1) QH$_2$ moves from Q$_{o1b}$ to Q$_{o1a}$, where the catalysis takes place. 2) The QH$_2$ oxidation starts by a rapid, exergonic PCET to the FeS centre. D309, which is unique to mycobacterial SC, stabilises the electron on the FeS centre see Fig. 6f slowing down the back transfer reaction. 3) The free energy from the initial PCET reaction is transduced to drive the second, endergonic PCET reaction, where the proton is transferred to D302/H110 via Tyr159, and the electron is transferred to heme $b_L$. 4) The electrons from the FeS centre and heme $b_L$ move to heme $c_1$ and heme $b_H$, respectively, and further along the two electron transfer branches to Complex IV and the Q$_i$ site. 5) The protons from the Q$_{o1a}$ site are released via proton wires that connect to the P-side bulk. Proton release from H110 downshifts the heme $b_L$ redox potential that could slow down the electron back-transfer from heme $b_H$. 6) The oxidised Q$_o$ is released into the membrane Q-pool, allowing for a new QH$_2$ to enter.

downshifted by ca. 100 mV and heme $b_L$ is upshifted by 50 mV relative to the WT model (Fig. 6f, Supplementary Table 5). Protonation of His355 increases the redox potential of the FeS Rieske centre by ca. 190 mV, leading to an effective proton-coupled redox potential of +120 mV (Fig. 6f). In this regard, Asp309 could help in establishing a downhill electron transfer between semiquinone and heme $b_L$, a process that is more favoured with the ubiquinol (+90 mV). We suggest that these adaptations could be important for the mycobacterial Complex III, where the Rieske domain does not undergo a conformational change that kinetically gates the electron transfer reaction[75]. The redox tuning effects around Q$_{o1a}$ could thus be critical for ensuring a high forward electron flux. Moreover, the quinone motion from Q$_{o1a}$ to Q$_{o1b}$ could provide additional entropic effects to compensate in part for the non-mobile Rieske protein.

The electron transfer between the quinone and heme $b_L$ is linked to a proton transfer from the semiquinone (QH·) to Asp302 (Fig. 5b, d), which is adjacent to His110. The His110/Asp302 ion-pair undergoes conformational changes in the MD simulations and could modulate the energetics of the proton release to the P-side bulk. Protonation of the His110/Asp302 also has a redox tuning effect on heme $b_L$, with the proton release to the P-side expected to downshift the redox potential of heme $b_L$ that could favour the forward electron transfer (Fig. 7).

The tuberculosis drug molecules, Q203 and TB47 interact with Asp309 (Glu314 in *M. tuberculosis*) and His355 (His375 in *M. tuberculosis*)[12,13] that could be further targeted in rational drug design, e.g. by introducing functional groups on the inhibitor that form specific interactions with these residues.

In the Q$_i$ site, we also observed changes relative to the canonical Complex III, specifically a π-stacking interaction between the menaquinone and Trp231, whereas Lys32 and/or Lys260 could serve as proton donors in the Q$_i$ reduction, catalysing proton uptake via the bound cardiolipin molecule (Fig. 3d, i, Supplementary Fig. 14) (cf. also refs. 30,63).

On the other side of the electron bifurcation pathway, our data suggested that the electrons are transferred from the Rieske FeS centre via the two heme $c$ cofactors of QcrC to Complex IV that form the rate-limiting step for the charge transfer due the large distance between the donor-acceptor sites, consistent with previous experiments[55,56]. By accounting for the experimental proton uptake rate[55], we could predict an overall turnover rate that is in excellent agreement with our experimental activity data. These findings support that the slow proton uptake in Complex IV module of the SC is rate-limiting for the overall long-range PCET process.

Our data also showed that the mycobacterial-specific SodC sub-unit is dynamically highly flexible, forming contacts with the QcrC domain, albeit the electron transfer from the Cu of SodC to the QcrC is expected to be rather slow based on the MD-sampled conformations. Although not supported by our current experimental data, it is interesting to speculate whether such putative electron transfer routes could provide an alternative electron input channel for the SC under specific conditions, and a detoxification mechanism for the mycobacterial SC to scavenge reactive oxygen species created, e.g. by human immune cells (cf. also refs. 10,76,77).

On the Complex IV side of the SC, the D-channel establishes a proton wire from Asp115 to Glu266, and further via a putative PLS near heme $a_3$ and to the P-side bulk (Fig. 4c). Glu266 sampled both upward and downward conformations in our MD simulations, enabling contacts with the active site/the PLS and the D-channel, whilst our high-resolution cryo-EM structure support only the downward conformations of the residue (but cf. refs. 15,41,42). Our MD simulations together with structural analysis[55] suggested that the D-channel is partially blocked by the QcrB loop (Fig. 4g), with several protonatable residues that could control the proton uptake into the channel (Supplementary Fig. 18a). The slow water exchange at this site (Supplementary Fig. 18b) could affect the proton uptake from the N-side and rationalise the pH-independent P → O transition in the *M. smegmatis* SC[55]. We suggest that the protonatable groups of the QcrB loop function as a proton-collecting antenna that renders the SC less sensitive to environmental changes. Our data also supported several water molecules in the K-channel that are involved in proton uptake to the active site. Despite

                    

key modular adaptations, the Complex IV domain of the SC is likely to be governed by overall similar functional principles as the canonical heme-copper oxidases, but establishing an overall rate-limiting step for the long-range charge transfer process. Moreover, the mycobacterial-specific residues and proton wires that control the primary PCET reactions in the $Q_o$ site of Complex III, provide a basis for site-directed mutagenesis experiments and establish principles for the rational drug design of inhibitors of mycobacterial SCs.

In conclusion, we have described here mechanistic principles of the obligate $III_2IV_2$ SC using large-scale molecular simulations in combination with structural experiments and activity measurements. Our simulations show that quinol can bind in the canonical $Q_{o1a}$ site, where it interacts with the nearby residues His355 and Tyr159. The binding mode is consistent with previous inhibitor-bound forms of the canonical Complex IIIs[46], whereas the unique $Q_{o1b}$ site is exchangeable with the former, and could provide a transient substrate docking site. Our QM/MM free energy calculations predict that the $Q_{o1a}$ site enables an exergonic PCET reaction from quinol to the FeS centre followed by a second, endergonic PCET towards the heme $b_L$. In this regard, modular adaptations of specific residues in the $Q_{o1a}$ site tune the energetics of these redox reactions, which could enable mycobacteria to use menaquinol as a substrate instead of ubiquinol and support electron transfer without the characteristic Rieske motion as in canonical Complex IIIs. Our MD simulations together with our structural data show functional proton wires in central sites of the SC, including the $Q_{o1a}$ site and $Q_i$ sites in Complex III, as well as around the active binuclear site and the D- and K-channels in Complex IV. Our predicted electron transfer rates in the SC are in good agreement with measured activities and further support that proton uptake via the D-channel is the rate-limiting step for the long-range charge transfer process. Taken together, our combined findings support previous data on the well-studied Complexes III and IV[24,26,32] but also highlight key differences in the quinone binding, proton pathways, and redox tuning principles that provide a basis for further mechanistic studies and for developing drug molecules against mycobacteria.

## Methods
### Molecular models
The $III_2IV_2$ supercomplex of *M. smegmatis* was modelled based on the cryo-EM models (PDB ID:6ADQ[11] and PDB ID: 6HWH[10]) (see Supplementary Fig. 1, Table 2). A SodC dimer was homology modelled based on a high-resolution x-ray structure of the protein from *Mycobacterium tuberculosis* (PDB ID; 1PZS[78]) using SWISS-model[79] and docked into the blurred cryo-EM density via the lipid membrane anchor, which we covalently linked to Cys21$^{SodC}$, as also observed in previous structures[10,11,80,81]. Similarly, Cys24 of subunit LpqE was also linked to a lipid membrane anchor. A menaquinol was modelled in three putative positions ($Q_{o1a}$, $Q_{o1b}$ and $Q_{o2}$). The $Q_{o1b}$ and $Q_{o2}$ sites were modelled based on the partial density observed in refs. 10,11,82, whilst the $Q_{o1a}$ position was modelled based on the resolved positions for quinone (PDB ID: 6Q9E) from ref. 16 and stigmatelin (PDB ID: 1PP9) from ref. 83 To this end, we applied a harmonic force ($k = 20$ kcal mol$^{-1}$ Å$^{-2}$) between the quinone in the $Q_{o1b}$ site and H355$^A$ and pulled the quinone gradually until it reached a hydrogen-bonding distance with H355$^A$ (3.5 Å between the nearest quinone oxygen and the NE2 of H355$^A$). During the minimisation, the protein backbone was fixed with residues only in QcrB allowed to relax. After optimisation of the initial position, the system was relaxed without restraints. Unresolved residues and sidechains that were also built into all MD models based on the cryo-EM density. The protein was modelled in both symmetric as well as asymmetric states, with regard to the opening angle of QcrC subunit (Supplementary Fig. 1), with a cross-correlation of 0.66 (EMDB: 9610) and 0.71 (EMDB: 0289) to respective cryo-EM maps for the resolved parts. The LpqE subunits were removed in the asymmetric model due to steric clashes (Supplementary Fig. 1), as also indirectly supported by

our biochemical assay (Fig. 6e). The MD models were embedded in a $266 \times 156$ Å 1-palmitoyl-2-oleoyl-*sn*-glycero-3-phosphocholine (POPC) membrane together with eight experimentally-resolved cardiolipin residues at the dimer interface. The MD models, comprising around 933,000–937,000 atoms, were further solvated with TIP3P water molecules and neutralised with around 783/665 Na$^+$/Cl$^-$ ions, comparable to a 150 mM NaCl concentration. Protein structure files were generated using CHARMM c38b[84].

### Molecular dynamics simulations
Atomistic MD simulations of the $III_2IV_2$ SC model were performed using the CHARMM36 force field for protein/lipids, ions, and water. Parameters for all cofactors were derived from in-house DFT calculations[85,86], with the remaining system treated using the CHARMM36m force field[87]. The MD simulations from the symmetric and asymmetric models with different menaquinone binding sites and oxidation states as well as all mutants are listed in Supplementary Table 1. Initial protonation states were obtained from using Poisson–Boltzmann electrostatic calculations with Monte Carlo sampling[88,89], based on models with the cofactors treated in their oxidised state (see Supplementary Table 3 and Table 15, for a list of residues with non-standard protonation states and their estimated p$K_a$ values, respectively). All MD simulations were performed in an *NPT* ensemble at $T = 310$ K and $p = 1$ atmosphere, using a 2 fs integration time step, and with electrostatics modelled using the particle mesh Ewald (PME) method. The system was gradually relaxed for 10 ns with harmonic restraints of 2 kcal mol$^{-1}$ Å$^{-1}$ on all protein and cofactor heavy atoms followed by 10 ns with harmonic restraints of 2 kcal mol$^{-1}$ Å$^{-1}$ on all protein backbone heavy atoms and by 10 ns equilibration with weak (0.5 kcal mol$^{-1}$ Å$^{-1}$) restraints on all $C_\alpha$ atoms, and finally ~0.5 μs production runs. All classical MD simulations were performed using NAMD2 (v. 2.12/2.13) for equilibration and NAMD3 (alpha9)[90] for production runs. The simulations were analysed using VMD 1.9.3/1.9.4[91] and MDAnalysis 2.0.0[92,93]. Hydration dynamics and central distances were analysed based with a frequency of ca. 2 frames/ns for each trajectory. Line plots show both raw data (shaded) as well as smoothed data (solid line) using a Savitzky-Golay filter, to improve readability.

### Derivation of force field parameters
Force field parameters for the redox cofactors were derived based on DFT models using a methodology described before[85,86,94–97]. To this end, the DFT models were constructed based on high-resolution crystal structures and optimised at the B3LYP-D3/def2-SVP(C,H,O,N)/def2-TZVP (Cu, Fe, S) level of theory[98–103] in different oxidation and/or ligand states (Supplementary Fig. 21). Force field parameters were derived for the lowest energy spin-state configuration, where applicable, based on the molecular Hessian[104,105], and adapted together with Lennard–Jones parameters for the CHARMM36 force field[87,106]. Atomic partial charges were calculated using the restrained electrostatic potential (RESP) method[107,108], based on single-point calculations at the B3LYP-D3/def2-TZVP level of theory. To this end, atomic partial charges were derived for quinone in the menaquinone (Q), menaquinol (QH$_2$, QH$^-$), and mena-semiquinone (QH$^\cdot$, Q$^{\cdot-}$) while bonding and Lennard–Jones parameters were derived from cgenff[86,109,110]. For 2His-coordinated heme $b$, and Met/His-ligated and Cys$_2$-coordinated heme $c$ in the Fe$^{II}$ and Fe$^{III}$ states both force constants and charges were derived based on DFT models. For the Rieske centre, the DFT models were built using the 2Fe2S-2Cys-2His core, truncated at $C_\alpha$ atoms, and modelled as ethyl groups, and optimised without restrains. The Rieske FeS cluster was optimised in the Fe$^{III}$Fe$^{III}$ and Fe$^{III}$Fe$^{II}$ states (cf. also refs. 86,96,97,103) as well as with the imidazole group modelled in both protonated and deprotonated states (Supplementary Fig. 21), using the broken-symmetry spin-flip approach[103] with anti-ferromagnetic coupling between the metal centres. The active site of SodC was modelled with the copper, in the oxidised state (Cu$^{II}$),

coordinated by a hydroxyl and three His residues, which were truncated at the $C_\beta$ atom, and kept fixed during the structure optimisation. Parameters for $Cu_A$ ($Cu^{II}Cu^I$ and $Cu^ICu^I$ forms); heme $a$ (in $Fe^{III}$ and $Fe^{II}$) heme $a_3$ and $Cu_B$ centre (in $Fe^{IV} = O^{2-}$, $Cu(II)-OH^-$ $TyrO^•$ / $Cu^{II}-OH^-$ $TyrO^-$ were adapted based on our previous work[85]. The redox cofactors included surrounding first solvation sphere ligands. The charge distribution of heme $a$ and heme $a_3$ thus contains the polarising effect of A/D propionates, as well as the Arg cluster above the propionates. All DFT models were optimised using TURBOMOLE v. 7.5.1[111]. The force field structure and dynamics of the cofactors were compared with results from QM/MM calculations. See Supplementary Fig. S21 for model systems used for parametrisation, and interaction validation as well as Supplementary Tables S7–S14 and ref. 85 for force field parameters. All parameters used for the MD simulation can be found on Zenodo (repository code: https://doi.org/10.5281/zenodo.10118429) and at https://github.com/KailaLab/ff_parameters.

## Electron transfer flux calculations

Electron transfer (eT) rates were estimated as described in ref. 112. Briefly, structures obtained from the MD simulations were used to obtain dynamically-averaged eT-rates, by considering all donor-acceptor distance pairs between the cofactors for the rate calculations. For estimating approximate coupling elements[113], we used the explicit intervening protein region via calculation of a protein coupling matrix between *all* individual eT-pathways. The approximate eT-rates were estimated based on the approximate electronic coupling elements based on the MD simulations, experimental redox potentials, and by using generic reorganisation energies of $\lambda = 0.7$ eV, except for the heme $a$/heme $a_3$ couple, where lower reorganisation energies of $\lambda = 0.3$–$0.5$ eV have been proposed[112,114,115]. The averaged eT-rates were computed based on the individual rates (ref. 112, cf. also ref. 113),

$$\log(k_{ij}) = 13 - (1.2 - 0.8\rho_{ij})\left(R_{ij}(\text{Å}^{-1}) - 3.6\right) - 3.1(\text{eV}^{-1})\frac{(\Delta G + \lambda)^2}{\lambda} \quad (1)$$

$$\langle k_{eT} \rangle = \frac{1}{N}\sum_{i,j}^{N} k_{i,j} \quad (2)$$

The $(1.2-0.8\rho_{ij})(R_{ij}-3.6)$ factor arises from the exponential dependence of the electronic coupling on the distance, $\exp(-\beta_o r)$, with the $\beta_o$ ranging between $2.8$ Å$^{-1}$ and $3.5$ Å$^{-1}$ in proteins[116,117], and calculated here as weighted sum of the explicit proton surroundings along all tunnelling pathways $N$ between atom pairs $i$, $j$ (see also Supplementary Fig. 12). The kinetic modelling of the long-range charge transfer was simulated using COPASI 4.35[118] with microscopic rates or transition barriers estimated from the eT rate calculations, QM/MM calculation for the PCET reactions, or from experiments (see Supplementary Fig. 12, Table 4).

Kinetic models were created for both the symmetric and asymmetric complexes. The first eT from $Q_{o1a}$ was restricted to the Rieske FeS, while the second one was restricted from $Q_{o1a}$ to heme $b_L$. The eT from $Q_{o1a}$ to heme $b_L$, and heme $a$ to heme $a_3$ were modelled as irreversible to account for $Q_{o1a,\,ox}$ unbinding and oxygen reduction, respectively, resulting in a non-equilibrium state that drives the overall reaction forward. In an alternative model, the rate of electron transfer to FeS and heme $b_L$ were obtained from the QM/MM calculations based on transition state theory, leading to similar results (Supplementary Fig. 13c). A second model was used to also consider the eT within the Complex III dimer in the asymmetric system. For the open QcrC state, with a disconnected eT chain, the Rieske FeS was assumed to be become fully reduced but blocked further eT from $Q_{o1a}$ along the cyt $cc$ chain. To this end, the second eT from $Q_{o1a}$ and the reduction of $Q_i$ were modelled as irreversible reactions, and the electron leakage from the open to closed side of the SC was modelled via eT between

heme $b_L$s cofactors of the Complex III dimer. On the closed side of the SC, reduction of heme $a_3$ was modelled as irreversible, whilst no bias was put on the eT direction in $Q_{o1a}$. To account for the proton-coupled electron transfer reaction in Complex IV, protonation on a pump site (PLS, see refs. 37,41,119) near the BNC modulated the $E_m$ value of the BNC, whereas the protonation rate was modelled as the experimentally determined 4 ms proton uptake via the D-channel, whereas the oxygen reduction was modelled as an irreversible reaction. The overall charge transfer reaction rates were calculated from the half-time ($t_{1/2}$) of heme $a_3$ reduction using the relationship $k = \ln 2/t_{1/2}$. See Data Availability for the kinetic models.

## Water cluster analysis

Clustering of water molecules in the MD simulations was performed using WATCLUST v0.1 plugin in VMD[120] with default parameters, except for "waternumbermin", which was set to ca. 10% of the frames, and a "dist" value of 0.3. The water clusters were coloured by the probability of observing the water molecules at the given positions.

## Water survival probability

Water survival probabilities at the entrance of the D-channel were calculated using the water dynamics module of MDAnalysis 2.0.0[92,93], selecting water molecules within 5 Å of Asp115 (bovine: D91) of subunit I in Complex IV. See ref. 39 for further details of the simulation setup of the bovine Complex IV simulations. Survival probabilities of water molecules in the D- and K-channel were calculated using a spherical volume with a 5 Å radius around residues Thr125 and Thr337.

## Channel volume estimation

Channel volumes of the D- and K-channel were estimated for the cryo-EM structure using CAVER3[121]. To this end, the channel search was initiated from Glu266/Thr182 and Tyr268/Thr337 with endpoints at Asp115 and Arg340 for the D- and K-channel, respectively. The residues at the endpoint were excluded for the channel search and analysed using probe radii of $0.85$–$0.9$ Å. The volume of the channel was estimated as a cylinder enclosing the region between start/endpoint and bulk, based on the channel length and average width.

## Redox potentials and binding energies from PBE calculations

Redox potentials and p$K_a$ values of cofactors and amino acids were estimated based on the continuum electrostatic calculations of the Poisson–Boltzmann equation, as implemented in APBS 1.5[122] in combination with Monte Carlo sampling of protonation and redox states, performed with Karlsberg+[88,89,123]. While these calculations consider thermodynamic effects, they do not account for how the substitutions affect reaction barriers or proton pathways linked to the PCET reactions. Charges for the different oxidation and protonation states were obtained similarly as before[85,86,124]. The protein was described using explicit atoms with atomic partial charges, embedded in an inhomogeneous dielectric continuum with a low dielectric constant of four. The bulk water was described by a homogeneous dielectric continuum with a dielectric constant of 80. The boundary interface between the protein and solvent was calculated by the molecular surface routine implemented in APBS, using a solvent probe radius of 1.4 Å, and modelling an implicit ionic strength with 100 mM potassium chloride. Protonation probabilities were probed along classical simulations (Supplementary Table 1) every 1 ns. A final MC round was performed considering only key residues and cofactors to improve convergence of the calculations.

Binding energies of menaquinone in different oxidation states ($QH_2$, $QH^-$, $QH^•$, $Q^•$ and $Q$) were estimated at the $Q_{o1a}$, $Q_{o1b}$ and $Q_{o2}$ sites in the Complex III dimer at the continuum electrostatic level using APBS 1.5 (PBSA-MM approximation). To this end, the protein was truncated to include subunits QcrA, QcrB, and QcrC and described with explicit point charges and a low dielectric constant of 4, and

extending 5 Å beyond the protein to account for the lipid membrane. The bulk water was described by a homogeneous dielectric continuum with a dielectric constant of 80. The boundary between protein and solvent was determined by the molecular surface routine implemented in APBS[122], using a solvent probe radius of 1.4 Å, with an implicit ionic strength with 100 mM potassium chloride. Binding energies of the quinone in the protein were computed using a thermodynamic cycle based on contributions of desolvation effects and electrostatic interactions of quinone species and the protein environment[125]. All calculations were based on the MD simulations of $QH_2$ in $Q_{o1a}$, $Q_{o1b}$ and $Q_{o2}$ (S2, S15, S16 in Table S1) by swapping the charges of menaquinone to the desired redox state, and re-optimising the hydrogen atom positions in Karlsberg+[88,89,123] (see above).

## QM/MM calculations

To study the PCET reactions linked to the electron bifurcation process at $Q_{o1a}$, two QM regions were defined comprising the quinol/Rieske centre part (QM1) and the semiquinone/heme $b_L$ part (QM2). The two PCET reactions in QM1 and QM2 were treated as sperate models at the QM/MM level. Hybrid QM/MM umbrella sampling (US) calculations were performed based on a structure extracted from an MD simulation of quinol obtained after 500 ns for the initial PCET step (QM1), and QM2 was constructed after the first PCET converged to the QH· state (Fig. 2A, B). The QM1 region (195 atoms) comprised the menaquinol headgroup (up to atom C10), the FeS Rieske centre coordinated by Cys333, Cys352, His335 and His355; and surrounding residues Leu336, Cys338, Cys354, Ser357, Phe156, Tyr159, Thr169, Ala174, Met305, Thr308, Asp309, Ile312, as well as 3 water molecules. The QM2 region (248 atoms) comprised the menaquinol (QH·) headgroup, heme $b_L$ with its coordinated His109 and His211, as well as Arg106, Tyr159, Asn282, Asp302, Tyr304, and eight water molecules. The MM region comprised the Complex III dimer, and a layer of water/lipids/ions around 5 Å of the protein, resulting in a model with around 85,000 atoms.

The electronic structure of the FeS Rieske centre was prepared by anti-ferromagnetically coupling the iron atoms using the broken-symmetry spin flip approach[126,127]. All QM calculations were performed at the B3LYP-D3[98,128,100] level with def2-SVP/def2-TZVP(Fe, S) basis sets. The QM regions were prepared with localised electrons[86] at the electron donor ($QH_2$ for the first reaction, and QH· for the reaction, see Supplementary Fig. 11e, f).

Link atoms, modelled as hydrogen atoms, were placed at the $C_\beta$–$C_\alpha$ bonds, with exception of arginine residues, where the link atoms were placed between the $C_\gamma$–$C_\delta$ bond and for menaquinone, where the link atom was placed between atom C9-C11 bond. To localise the diabatic electron transfer state, the system was relaxed separately in their corresponding redox states, and the molecular orbitals were merged with the subsystems[86]. The QM/MM systems were relaxed using an adopted basis Newton-Raphson optimiser until the energy (0.0006 kcal mol$^{-1}$) and gradient convergence threshold (0.0006 kcal mol$^{-1}$ Å$^{-1}$) were reached.

Potential energy profiles for the PCET reactions were computed based on reaction pathway optimisation for the proton transfer reaction coordinate, constrained with a harmonic potential using a force constant of 500 kcal mol$^{-1}$ Å$^{-1}$, modelled as a linear combination of bond-forming and bond-breaking distances (Supplementary Fig. 11a, b). QM/MM umbrella sampling (QM/MM-US) simulations were initiated from the intermediate states along the PCET reactions using a force constant of 100–500 kcal mol$^{-1}$ Å$^{-2}$ applied on the proton transfer reaction coordinate (see Supplementary Fig. 11c, d), with the QM/MM US simulations divided amongst 60 windows. The potential of mean force was computed using the weighted histogram analysis method (WHAM) with a converge threshold set to 0.0001 kcal mol$^{-1}$, and statistical errors were estimated using bootstrapping analysis.

The QM/MM MD simulations were performed at $T$ = 310 K using a 1 fs integration time step, with a 12 Å sphere around the QM region that was allowed to relax during the calculations. All QM/MM calculations were performed using an in-house version of the CHARMM c38b/TURBOMOLE 7.5.1 python wrapper[129]. All DFT calculations were performed using TURBOMOLE v.7.5.1[111], and the electronic structure was visualised using VMD 1.9.3/1.9.4[91].

To study proton donors at the $Q_i$ site, QM/MM MD simulations were performed based on the last frame from a 0.5 µs classical MD simulation (simulation S6, Supplementary Table 1). The QM region comprised the menaquinol head group, Gln29, Lys32, Glu44, Tyr 48, Trp231, Phe257, Lys260, Ser261, 15 water molecules and one POPC giving a total of 299 atoms and a net charge of zero. The system was minimised 15 Å around the QM region, following the reduction of the Q before starting the QM/MM MD simulations, with the MM region comprising 78,346 atoms. Link atoms around the $Q_i$ site were placed between $C_\beta$–$C_\alpha$ atoms with exception of menaquinone, with a link atom placed between the C9–C11 bond. All QM calculations were performed at the B3LYP-D3[98,128,100] level with def2-SVP basis sets. The QM/MM MD simulations were performed at $T$ = 310 K using a 1 fs integration time step, and by fixing heavy atoms around 15 Å of the QM region to avoid exchange of QM and MM water molecules. The simulations were performed for 1 ps in three replicas.

To investigate possible BNC ligands in the continuous cryo-EM density, we performed QM/MM optimisations of the BNC where the QM-regions comprised heme $a_3$, His397, $Cu_B$, His264, His313, His314, Tyr268, and two water molecules (ca. 140 atoms). In this regard, we optimised the following states: $Fe^{III}$–$OH^-$–$Cu^{II}$/$TyrO^-$, $Fe^{III}$–$H_2O$–$Cu^{II}$/$TyrO^-$, $Fe^{III}$–$OOH^-$–$Cu^{II}$/$TyrO^-$, $Fe^{III}$–$OH_2$/$HO$–$Cu^{II}$/$TyrO^-$, $Fe^{III}$–$OH^-$/$H_2O$–$Cu^{II}$/$TyrO^-$, $Fe^{III}$–$OH^-$/$HO$–$Cu^{II}$/$TyrO^-$ and $Fe^{III}$–$OH^-$–$H_2O$–$Cu^{II}$/$TyrO^-$ (see Fig. 4h, Supplementary Fig. 7e-f) at the B3LYP-D3/ def2-SVP/ def2-TZVP(Cu, Fe) level of theory with the MM region composed of remaining Complex IV system.

## Cell growth

*M. smegmatis* mc²155 cells with the SC FLAG-tagged on QcrB, were grown in 7H9 medium, supplemented with ADS (5 g L$^{-1}$ BSA, 2 g L$^{-1}$ dextrose, and 0.8 g L$^{-1}$ NaCl), 25 µg mL$^{-1}$ kanamycin, 50 µg mL$^{-1}$ hygromycin, 0.2% glycerol, and 0.05% Tween 80. Single colonies were picked from 7H9 agar plates supplemented with ADS and antibiotics, inoculated into 25 mL culture, and shaken at 180 rpm, 30 °C. The pre-cultures were diluted after 48 h into 1000 mL 7H9 medium in a 2 L flask, and shaken at 180 rpm at 30 °C for ~72 h. Cells were harvested after $OD_{600}$ reached 2.5.

## Membrane preparation

Cells were homogenised in cell lysis buffer (50 mM Tris-HCl, pH 7.5, 50 mM NaCl, 0.5 mM EDTA) in the presence of phenylmethanesulfonyl fluoride and DNase I (Roche), and crushed with a cell disrupter (Constant Systems) with four cycles at 35 kPsi. Cell debris was then removed by centrifugation at 18,600 g for 30 min, followed by collection of the membranes after ultracentrifugation at 147,000 g for 90 min.

## Isolation of the supercomplex

Membranes (1 g) were incubated in 10 mL of solubilisation buffer (50 mM Tris-HCl, pH 7.5, 100 mM NaCl, 0.5 mM EDTA, 2% (w/v) GDN or 1% (w/v) n-dodecyl β-d-maltoside (DDM) (Anatrace) and incubated overnight or 45 min, respectively, at 4 °C under stirring. Unsolubilised material was removed by ultracentrifugation at 147,000 g for 30 min. The supernatant was then applied to an anti-FLAG M2 column (1 mL bed volume, Sigma Aldrich). The column was washed with four column volumes of washing buffer (50 mM Tris-HCl, pH 7.5, 100 mM NaCl, 0.5 mM EDTA, 0.01% (w/v) GDN or 0.05% (w/v) DDM). Protein was eluted with 5 column volumes of elution buffer (washing buffer +5 mg/mL FLAG peptide (Sigma Aldrich)) and concentrated with a 100 kDa molecular weight cut-off concentrator (Merck

Millipore). The protein samples were aliquoted, flash-frozen in liquid $N_2$ and stored at −80 °C.

## Activity assays

The menaquinol oxidation: $O_2$ reduction activity of the purified *M. smegmatis* SC was measured by following the $O_2$ reduction rate upon addition of reduced substrate; 2,3-dimethyl-[1,4] naphthoquinone (Rare Chemicals GmbH) as described in refs. 22,55. The activity was measured in a buffer solution (50 mM Tris-HCl, pH 7.5, 100 mM NaCl, 0.5 mM EDTA, 0.01% (w/v) GDN or 0.05% (w/v) DDM at 25 °C) using a Clark-type oxygen electrode (Hansatech instruments). The reaction was started by addition of 5 μL of 20 mM substrate solution into an electrode chamber containing 1 mL buffered protein solution and 500 nM bovine Sod, in order to reduce the effect of the substrate auto-oxidation (cf. ref. 12). The activity was obtained from the initial slope of the graph where the $O_2$ concentration was linearly dependent on time. Background $O_2$ reduction rate in the absence of the SC was measured as a control and subtracted from that measured in the presence of the SC. The activity was calculated using the following equation:

$$Activity\left[e^- s^{-1}\right] = \frac{c_{ox} \cdot 4}{60 \cdot c_{prot}} \quad (3)$$

where $c_{ox}$ is the change in oxygen concentration over one minute, 4 is the number of electrons needed to reduce oxygen to water and $c_{prot}$ is the concentration of the protein. The protein concentration was determined from difference spectra of the dithionite-reduced minus oxidised states of the SC, recorded with a spectrophotometer (Cary 100, Agilent Technologies), and calculated using the given absorption coefficients: $\varepsilon_{605-630\,nm} = 24$ mM$^{-1}$cm$^{-1}$ (heme *a*), $\varepsilon_{562-577\,nm} = 22$ mM$^{-1}$cm$^{-1}$ (heme *b*) and $\varepsilon_{552-540\,nm} = 19$ mM$^{-1}$cm$^{-1}$ (heme *c*).

## Cryo-EM grid preparation and data collection

Purified protein sample of supercomplex (3.5 mg mL$^{-1}$) was incubated with substrate (2,3-dimethyl-[1,4] naphthoquinone, 100 μM) and inhibitor (lansoprazole sulphide, Tokyo Chemical Industry, Japan; 0.5 mM) for 45 min at room temperature. After incubation, 3 μL of sample were applied to holey carbon film coated copper EM grids (300 mesh R2/2 grid, Quantifoil, Micro Tools GmbH, Germany), blotted for 3 s at 4 °C (100% humidity) and plunge-frozen in liquid ethane using Vitrobot Mark VI (Thermo Fisher Scientific). The grids were glow-discharged in air at 20 mA for 120 s (PELCO easiGlow) before sample application. Data collection was performed at 300 kV using Titan Krios G3i electron microscope (Thermo Fisher Scientific) equipped with K3 Gatan detector. Dataset 1 (2.8 Å) of 18164 exposures (40 exposure fractions each) was acquired in electron-counting mode at a nominal magnification of 105000 (0.846 Å/pixel). Automated data collection was set up using EPU software package (v2.12.1; Thermo Fisher Scientific). Camera exposure rate was 17.6 e$^-$/pixel/s and total exposure of the sample was 48.2 e$^-$/Å$^2$ (Supplementary Table 6). Dataset 2 (2.3 Å) of 24868 exposures (40 exposure fractions each) was acquired in electron-counting mode at a nominal magnification of 105000 (0.828 Å/pixel). Automated data collection was set up using EPU software package (v2.12.1). Camera exposure rate was 17.5 e$^-$/pixel/s and total exposure of the sample was 40 e$^-$/Å$^2$ (Supplementary Table 6).

## Data processing, model building and refinement

The cryo-EM data was analysed using cryoSPARC v3[130] (dataset 1) and cryoSPARC v4 (dataset 2). with motion correction and CTF estimation performed using patch motion correction and CTFFIND4 (dataset 1) or patch CTF estimation (dataset 2), respectively. For the dataset 1, templates for the particle selection were generated by manual particle picking and 2D classification, resulting in 752,267 selected particles.

The number of particle images was reduced by 2D classification to 295,370 and by further ab initio reconstruction leading to 90,918 particles. The dataset was further refined using homogenous refinement, followed by local CTF and non-uniform refinement[131] without enforcing symmetry, resulting in a final map with an overall resolution of 2.8 Å (Supplementary Fig. 16). Templates generated for the dataset 2 were picked from over 5 million particles, with ca. 800,000 which were selected by the first round of 2D classification. Both datasets show a symmetric supercomplex and feature both LpqE and SodC. Initial structural models were built based on previously refined structures[10,11] (PDB ID: 6HWH[10] and PDB ID: 6ADQ[11]), providing a reasonable starting model for the refinement, followed by fitting into the map and manual adjustment using Coot 0.8.9.2[132]. The models contain in addition to canonical redox cofactors, ten menaquinone molecules, 249/614 buried water molecules, and 21/38 lipid molecules (Fig. 1b, Supplementary Figs. 17). The model also contains six short additional chains on the N-side of the complex, four of which were modelled as poly-alanine residues due to unresolved sidechains. Model refinement was performed using real-space refinement in Phenix 1.118.2[133] and Refmac 5.8.0257 from CCP-EM[134] as well as MD flexible fitting as implemented in NAMD2.13, with manual adjustments in Coot. The structural features of the model were validated using MolProbity[135]. Examples densities, rendered in UCSF ChimeraX 1.5[136], are shown in Figs. 1,3,4 and Supplementary Figs. 7, 10, 17, with map thresholds given based on the map level in UCSF ChimeraX. The refined structure and density are deposited with access codes: PDB ID: 8OVD and 8OVC and EMBD: 17211 and 17,210 for the 2.3 Å and 2.8 Å maps, respectively.

## Reporting summary

Further information on research design is available in the Nature Portfolio Reporting Summary linked to this article.

# Data availability

The data that support this study are available from the corresponding authors upon request. Cryo-EM maps are available in the Electron Microscopy Data Bank (EMD−17211 [https://www.ebi.ac.uk/pdbe/entry/emdb/EMD-17211] (III$_2$IV$_2$-SC, 2.3 Å map) and EMD-17210 (III$_2$IV$_2$-SC, 2.8 Å map)), and atomic models of the *M. smegmatis* supercomplex are available in the Protein Data Bank (PDB − 8OVD (III$_2$IV$_2$-SC, 2.3 Å model) and 8OVC (III$_2$IV$_2$-SC, 2.8 Å model)). Previous structures, referred to in the manuscript, can found on the Protein Data Bank with the following accession codes: 6HWH (III$_2$IV$_2$-SC), 6ADQ (III$_2$IV$_2$-SC), 7RH5 (III$_2$IV$_2$-SC), 7RH6 (III$_2$IV$_2$-SC), 7RH7 (III$_2$IV$_2$-SC), 7E1V (III$_2$IV$_2$-SC), 7QHO (III$_2$IV$_2$-SC), 7COH (Complex IV), 7ATE (Complex IV), 7AU6 (Complex IV), 6Q9E (Complex III), 1PP9 (Complex III), 1KYO (Complex III), 1PZS (SOD). Cryo-EM maps from previous structures used to calculate cross-correlation can be access in the Electron Microscopy Data Bank (EMD−0289 [https://www.ebi.ac.uk/pdbe/entry/emdb/EMD-0289] (III$_2$IV$_2$-SC, 6HWH) and EMD-9610 (III$_2$IV$_2$-SC, 6ADQ)). Models, parameters, and key trajectories have been deposited to the Zenodo database [https://doi.org/10.5281/zenodo.10118429].

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

## Acknowledgements

The project was supported by the Knut and Alice Wallenberg Foundation (KAW 2019.0043 to VRIK, MH, and PB, and KAW 2019.0251 to VRIK) and the Swedish Research Council. This work was also supported by the Swedish National Infrastructure for Computing (SNIC/NAISS 2022/1-29, 2021/1-40, 2020/1-38 to VRIK) at Centre for High Performance Computing (PDC) Centre, partially funded by the Swedish Research Council through Grant Agreement 016-07213, as well as LUMI-Sweden (project: 2022/13-14 to VRIK), and the Leibniz Rechenzentrum (LRZ, project:pr83ro to VRIK). The cryo-EM data were collected at the Cryo-EM Swedish National Facility funded by the Knut and Alice Wallenberg, Family Erling Persson and Kempe Foundations, SciLifeLab, Stockholm University and Umeå University.

## Author contributions

V.R.I.K. designed the study; D.R., A.P.G.H., S.L.M., V.R.I.K performed molecular simulations; T.K. prepared cryo-EM grids, collected cryo-EM data, processed cryo-EM data; T.K., D.R., A.P.G.H., V.R.I.K. built cryo-EM models; T.K., D.R., A.P.G.H., V.R.I.K. analysed and interpreted the cryo-EM models; S.M.K. isolated and biochemically characterised the protein; D.R., A.P.G.H., T.K., S.M.K., S.L.M., D.S., M.H., P.B., V.R.I.K. analysed the data; D.R., A.P.G.H. developed new analytical tools; V.R.I.K. directed the project; V.R.I.K., D.R. and A.P.G.H. wrote the manuscript with contribution from all authors.

## Funding

## Competing interests

The authors declare no competing interests.
