## [Peer Review File · Nature Communications]

Long-Range Charge Transfer Mechanism of the III₂IV₂ Mycobacterial SupercomplexEditorial Note: This manuscript has been previously reviewed at another journal that is not operating a transparent peer review scheme. This document only contains reviewer comments and rebuttal letters for versions considered at *Nature Communications*.

Reviewer #1 (Remarks to the Author):

In their revised manuscript, Riepl et al have thoroughly addressed the three reviewers' comments and my personal opinion is that the new manuscript is improved.

Below are some further comments that the authors may want to consider (in order of their appearance in the text rather than importance):

1. Introduction p2, 4th paragraph. 'The SC also features a C-type superoxide dismutase (SodC) and the accessory subunit LpqE'. Considering that LpqE is essential to activity, the author may want to remove 'accessory' in that sentence.
2. p2-3, I would still encourage the authors to explicitly define their use of 'canonical' as being those complexes III or IV which are not in obligate SC.
3. p2-3-4 'distinctly different mechanistic principles' and 'elusive' are maybe too strong descriptors. If anything in that system, things look much more straightforward than in the 'canonical' complexes. All the redox centres are fixed and lined up, eT are governed by differences in midpoint potentials and there are clear candidates for proton uptake. The work is of high quality and the kinetic model is informative but there is no surprise in the findings.
4. p8, 'Purification of the SC with n-dodecyl- β -D-maltoside (DDM) leads to loss of the LpqE subunit (Fig. 6e, see also Ref.12), and around 30% reduction of the SC activity (66 s⁻¹) relative to the GDN-solubilized SC, where all subunits are present (Fig. 6c).' I still don't think that the gel shown in fig 6e shows loss of LpqE and that it is wrongly referred to in the text. Instead it would be better referred to to explain why the reduction in activity is of only 30%: because fig 6e shows that there is still a good amount of LpqE in the sample.
5. p12, typo, 'resides' should be 'residues'?
6. Details of method used to determine protein concentration in the sample for activity calculation are still lacking.

Reviewer #2 (Remarks to the Author):

The manuscript aims at addressing a general issue of long-range electron transfer and associated proton transfers in enzymes of energy conserving systems, specifically complexes III and IV of respiratory chains. The approach is based on structural analysis with a strong input from MD simulations. While I admit that overall the addressed subject is very important, I cannot help the feeling that the manuscript is written in a chaotic manner and lacks clearly formulated goal/goals and specific conclusions that would enrich the already available research on the functioning of complexes III and IV. It contains many mental abbreviations that are incomprehensible to the reader, the figures presented are difficult to read and lack basic information, such as the distribution of hydrogen bonds, which is necessary to determine whether the binding site of menaquinol is not accidental. The work contains the results that are potentially attractive for publication, but in my opinion they should be carefully re-considered, supplemented and presented in the context of the specific problem that was solved. In its current form, the article is not suitable for publication.

To be more specific, below I list some of the major concerns and inconsistencies:

- 1) The Qo2 and Qo1b non-canonical binding sites of menaquinol were obtained from cryo-EM experiment. However, it is not mentioned how the canonical binding site (Qo1a) was estimated. The

results of MD simulations presented by the Authors revealed that the binding of menaquinol at the Qo1a is weak and unstable, which may indicate that it is not properly agreed. (page 6: " Interestingly, in some simulations initiated from the Qo1a site, we observe a transient motion of the quinol towards the Qo1b site, suggesting that these regions are dynamically exchangeable, but we could not observe substrate tunnels connecting the accessory Qo2 site with the Qo1a/b sites.").

2) Page 6: "In this regard, the Qo1a menaquinol is stabilised by His355 and Asp309, and water-mediated contacts with Tyr159, which in turn forms contacts with Asp302 of the characteristic PDWY motif 46,47 (Fig. 2b, 3a,b, Supplementary Fig. 9). Asp302 establishes an ion-pair with His110, the conformation of which modulates the binding affinity of the substrate (Supplementary Fig. 8c)." - The described interactions should be shown in at least one figure, but unfortunately they are not in any of the mentioned ones. Moreover in Fig. 2b one hydroxyl group of menaquinol seems to interact with H355 and possibly with D309, which may accept the first proton, but according the figure, the second hydroxyl group is directed towards the F156, which rather questions the correct conformation of menaquinol at the Qo site. In Fig. 3b the water occupancies of QH2 nad Qox are presented in graph, thus this figure is not a good reference for presenting binding mode of menaquinol. An unexplained abbreviation Qox was introduced in the drawing caption of Fig. 3b - is it quinone?

3) Fig. 5 presents energy profiles for two steps of PCET reaction occurring at the Qo site. The barriers to these processes seem reasonable, but what is puzzling is the high instability of quinone compared to semiquinone and quinol. The text lacks an explanation for this result because if both quinol and semiquinone are more stable, then the formation of quinone is thermodynamically unfavorable. In the Extended methods/molecular models subsection Authors mentioned that MD simulations were performed for three possible position of menaquinol. The high energy of menaquinone obtained from QM/MM calculations indicates that binding manner of menaquinone might be even more interesting and worth testing.

4) Page 8: "On the acceptor side of Complex III, we observe a subtle motion of the Qi quinone from the resolved binding position to a site, ca. 4 Å further from heme bL , where the quinone forms a π -stacking interaction with Trp231 in the MD simulations (Fig. 3d, e). The heme bL - heme bH - Qi network favours rapid electron transfer (103 -104 s⁻¹ , Fig. 6a), with our QM/MM simulations (Supplementary Fig. 14) further suggesting that Lys260 or Lys32 could function as local proton donors upon the quinone reduction (Supplementary Fig. 14, Movie 3), following uptake of the protons from the N-side via the nearby cardiolipin gate (see below, and cf. also Refs. 30,62)." - Figure 3d is illegible, the mentioned lysines K260 and K32 seem to be quite distant from the quinone carbonyl groups as for proton donors - it is necessary to present hydrogen bonds here. Fig 14 in Supplementary shows transport of a proton from K260 via two water molecules, while K32 appears to be quite far from the second carbonyl group of the substrate. The figure shows the transport of a proton onto the Q2⁻ ion, which is very unstable and such transport will certainly occur. Was it possible to obtain a semiquinone for this model, after all, in one Qo cycle one electron is transferred to Qi site. It is also necessary to present the energetics of the studied processes.

4) It is difficult to verified parameters used in MD simulations. Methods used are presented, but there is a lack of information on the structure of the models used to parameterize the redox cofactors. The figures presenting such models should be included in the Supplementary. This is important since in the case of FeS clusters, the protein surrounding significantly affects the cofactor parameters, its geometry and partial charges, while in the case of hemes, taking into account propionate groups without taking into account the strong electrostatic interactions they form with protein also distorts their bonding and nonbonding parameters, and thus affects the MD results. Correct parameterization of redox cofactors in the cases of the considered reactions in which they influence the substrate binding and what is more important directly participate in catalysis is crucial to obtain reliable results of MD simulation.

5) The article lacks discussion on the protonation state of residues directly interacting with quinol/quinone and their impact on the binding of the reactant and the catalytic abilities of the enzyme. Were the protonation states of H355, D302, D309, H110 verified in any way? What are the pKa values of these residues? A similar question arises about protein residues interacting with the quinone at the Qi site.

Answers to Reviewer #1 (Remarks to the Author):

Comment: In their revised manuscript, Riepl et al have thoroughly addressed the three reviewers' comments and my personal opinion is that the new manuscript is improved.

Answer: We would like to thank the Reviewer again for the insightful comments that have helped us to improve our manuscript.

1. Comment: 1. Introduction p2, 4th paragraph. 'The SC also features a C-type superoxide dismutase (SodC) and the accessory subunit LpqE'. Considering that LpqE is essential to activity, the author may want to remove 'accessory' in that sentence.

Answer: We agree that the term "accessory" underemphasizes the importance of the LpqE subunit and have thus remove it.

Revisions in the main text:

The SC also features a C-type superoxide dismutase (SodC) and subunit LpqE, both of which are anchored to the membrane by lipid modifications (Supplementary Fig. 1c, d).

2. Comment: p2-3, I would still encourage the authors to explicitly define their use of 'canonical' as being those complexes III or IV which are not in obligate SC.

Answer: We have now clarified the term canonical following the Reviewer's suggestion.

Revisions in the main text:

Despite the structural changes that are likely to tune the energetics of the charge transfer process (see below), the conserved structural elements⁵ of the mycobacterial SC, suggest that it could utilise overall similar charge transfer pathways as Complex III and IV^{1,24-26} variants, not found in obligate bacterial SCs (from here on referred to as canonical Complexes III and IV, Fig. 1).

1. Kaila, V.R.I. & Wikström, M. Architecture of bacterial respiratory chains. *Nat. Rev. Microbiol.* **19**, 319-330 (2021).

5. Brzezinski, P., Moe, A. & Ådelroth, P. Structure and Mechanism of Respiratory III-IV Supercomplexes in Bioenergetic Membranes. *Chem. Rev.* (2021).

24. Kaila, V.R.I., Verkhovsky, M.I. & Wikström, M. Proton-Coupled Electron Transfer in Cytochrome c Oxidase. *Chem. Rev.* **110**, 7062-7081 (2010).

25. Wikström, M. et al. New Perspectives on Proton Pumping in Cellular Respiration. *Chem. Rev.* **115**, 2196-2221 (2015).

26. Crofts, A.R. The modified Q-cycle: A look back at its development and forward to a functional model. *Biochim. Biophys. Acta Bioenerg.* **1862**, 148417 (2021).

3. Comment: p2-3-4 'distinctly different mechanistic principles' and 'elusive' are maybe too strong descriptors. If anything in that system, things look much more straightforward than in the 'canonical' complexes. All the redox centres are fixed and lined up, eT are governed by differences in midpoint potentials and there are clear candidates for proton uptake. The work is of high quality and the kinetic model is informative but there is no surprise in the findings.

Answer: We agree with the Reviewer and have removed the formulations. The obligate supercomplex does indeed seem in some regards more straightforward than the canonical

complexes. We thank Reviwer for assessing that our work is of high quality. We have revised the sentence here accordingly.

Revisions in the main text:

Despite the structural changes that are likely to tune the energetics of the charge transfer process (see below), the conserved structural elements⁵ of the mycobacterial SC, suggest that it could utilise overall similar charge transfer pathways as Complex III and IV^{1,24-26} variants, not found in obligate bacterial SCs (from here on referred to as canonical Complexes III and IV, Fig. 1).

*To address the functional dynamics responsible for **this fascinating** long-range charge transfer process in the mycobacterial III₂IV₂ SC, we integrate here structural, functional, and computational methods.*

1. Kaila, V.R.I. & Wikström, M. Architecture of bacterial respiratory chains. *Nat. Rev. Microbiol.* **19**, 319-330 (2021).
5. Brzezinski, P., Moe, A. & Ådelroth, P. Structure and Mechanism of Respiratory III–IV Supercomplexes in Bioenergetic Membranes. *Chem. Rev.* (2021).
24. Kaila, V.R.I., Verkhovskiy, M.I. & Wikström, M. Proton-Coupled Electron Transfer in Cytochrome c Oxidase. *Chem. Rev.* **110**, 7062-7081 (2010).
25. Wikström, M. et al. New Perspectives on Proton Pumping in Cellular Respiration. *Chem. Rev.* **115**, 2196-2221 (2015).
26. Crofts, A.R. The modified Q-cycle: A look back at its development and forward to a functional model. *Biochim. Biophys. Acta Bioenerg.* **1862**, 148417 (2021).

4. Comment: p8, 'Purification of the SC with n-dodecyl-β-D-maltoside (DDM) leads to loss of the LpqE subunit (Fig. 6e, see also Ref.12), and around 30% reduction of the SC activity (66 s⁻¹) relative to the GDN-solubilized SC, where all subunits are present (Fig. 6c).' I still don't think that the gel shown in fig 6e shows loss of LpqE and that it is wrongly referred to in the text. Instead it would be better referred to to explain why the reduction in activity is of only 30%: because fig 6e shows that there is still a good amount of LpqE in the sample.

Answer: We thank the Reviewer for pointing out that further clarification was needed. To this end, we have now specified that dissociation of LpqE in the DDM preparation is supported by structural data, but that some LpqE is still in the sample or even bound to the SC, which could explain that the activity is only reduced by 30%.

Revisions in the main text:

*Purification of the SC with n-dodecyl-β-D-maltoside (DDM) leads to **dissociation of LpqE from the SC, which is supported by structural data showing only a weak density for the subunit. Small amounts of the LpqE subunit remain in the sample or associated to the SC (Fig. 6e, see also Ref.12), and could explain why the DDM preparation shows only** around 30% reduction of the SC activity (66 s⁻¹) relative to the GDN-solubilized SC, where all subunits are present (Fig. 6c), **instead of a 50% reduction as may be expected, if one electron transfer branch would be completely blocked.***

12. Yanofsky, D.J. et al. Structure of mycobacterial CIII₂CIV₂ respiratory supercomplex bound to the tuberculosis drug candidate telacebec (Q203). *eLife* **10**, e71959. (2021).

5. Comment: 12, typo, 'resides' should be 'residues'?

Answer: We have fixed the typo.

Revisions in the main text:

Our MD simulations together with structural analysis⁵⁴ suggested that the D-channel is partially blocked by the QcrB loop (Fig. 4g), with several protonatable residues that could control the proton uptake into the channel (Supplementary Fig. 18a).

54. Yuly, J.L. et al. Electron bifurcation: progress and grand challenges. *Chem. Commun.* **55**, 11823-11832 (2019).

6. Comment: Details of method used to determine protein concentration in the sample for activity calculation are still lacking.

Answer: We have now clarified how the protein concentration is determined.

Addition to the extended methods:

The protein concentration was determined from difference spectra of the dithionite-reduced minus oxidised states of the SC, recorded with a spectrophotometer (Cary 100, Agilent Technologies), and calculated using the given absorption coefficients: $\epsilon_{605-630\text{ nm}} = 24\text{ mM}^{-1}\text{cm}^{-1}$ (heme a), $\epsilon_{562-577\text{ nm}} = 22\text{ mM}^{-1}\text{cm}^{-1}$ (heme b) and $\epsilon_{552-540\text{ nm}} = 19\text{ mM}^{-1}\text{cm}^{-1}$ (heme c).

Answers to comments by Reviewer #2 (Remarks to the Author):

Comment: The manuscript aims at addressing a general issue of long-range electron transfer and associated proton transfers in enzymes of energy conserving systems, specifically complexes III and IV of respiratory chains. The approach is based on structural analysis with a strong input from MD simulations. While I admit that overall the addressed subject is very important, I cannot help the feeling that the manuscript is written in a chaotic manner and lacks clearly formulated goal/goals and specific conclusions that would enrich the already available research on the functioning of complexes III and IV. It contains many mental abbreviations that are incomprehensible to the reader, the figures presented are difficult to read and lack basic information, such as the distribution of hydrogen bonds, which is necessary to determine whether the binding site of menaquinol is not accidental. The work contains the results that are potentially attractive for publication, but in my opinion they should be carefully re-considered, supplemented and presented in the context of the specific problem that was solved. In its current form, the article is not suitable for publication.

Answer: We thank the Reviewer for his/her comments and further suggestions. We have carefully addressed each point by improving figures, and clarifying central goals and their relation to the previous data on Complexes III and IV to improve the manuscript. We have attempted to clarify the discussion in the main text by re-formulating parts of the complete text, and re-ordered sections to make the presentation more logical. Answers to specific comments can be found below.

To be more specific, below I list some of the major concerns and inconsistencies:

Question 1: The Qo2 and Qo1b non-canonical binding sites of menaquinol were obtained from cryo-EM experiment. However, it is not mentioned how the canonical binding site (Qo1a) was estimated. The results of MD simulations presented by the Authors revealed that the binding of menaquinol at the Qo1a is weak and unstable, which may indicate that it is not properly agreed.

(page 6: " Interestingly, in some simulations initiated from the Qo1a site, we observe a transient motion of the quinol towards the Qo1b site, suggesting that these regions are dynamically exchangeable, but we could not observe substrate tunnels connecting the accessory Qo2 site with the Qo1a/b sites.").

Answer: We have now explained in the Extended Methods section that the position of quinone in Q_{o1a} was modelled based on comparison with canonical Complex III structures, where a ubiquinone molecule was previously observed (Ref. 8), and where inhibitors were identified in one of the crystal structures (Ref. 9). We find that the menaquinol in Q_{o1a} is stabilised in the narrow tunnel by hydrogen-bonds with His355 and Tyr159 in the MD simulations (Fig 2b). Binding of the quinol/quinone is enthalpically stabilised by a 3-4 kcal mol⁻¹ in both Q_{o1a} and Q_{o1b}, which is comparable to the experimentally estimated apparent K_m of 120 μM (Ref. 12).

8. Letts, J.A. *et al.* Structures of Respiratory Supercomplex I+III2 Reveal Functional and Conformational Crosstalk. *Mol Cell* **75**, 1131-1146 e6 (2019).
9. Huang, L.-s., Cobessi, D., Tung, E.Y. & Berry, E.A. Binding of the Respiratory Chain Inhibitor Antimycin to the Mitochondrial bc₁ Complex: A New Crystal Structure Reveals an Altered Intramolecular Hydrogen-bonding Pattern. *J. Mol. Biol.* **351**, 573-597 (2005).
12. Gong, H. *et al.* An electron transfer path connects subunits of a mycobacterial respiratory supercomplex. *Science* **362**, eaat8923 (2018).

Additions to the extended methods:

A menaquinol was modelled in three putative positions (Q_{o1a} , Q_{o1b} and Q_{o2}). The Q_{o1b} and Q_{o2} sites were modelled based on the partial density observed in Ref.^{1,2,7}, whilst the Q_{o1a} position was modelled based on the resolved positions for quinone (PDB ID: 6Q9E) from Ref.⁸ and stigmatellin (PDB ID: 1PP9) from Ref.⁹ To this end, we applied a harmonic force ($k= 20 \text{ kcal mol}^{-1} \text{ \AA}^{-2}$) between the quinone in the Q_{o1b} site and H355^A, and pulled the quinone gradually until it reached a hydrogen-bonding distance with H355^A (3.5 Å between the nearest quinone oxygen and the NE2 of H355^A). During the minimisation, the protein backbone was fixed with residues only in QcrB allowed to relax. After optimisation of the initial position, the system was relaxed without restraints.

1. Gong, H. *et al.* An electron transfer path connects subunits of a mycobacterial respiratory supercomplex. *Science* **362**, eaat8923 (2018).
2. Wiseman, B. *et al.* Structure of a functional obligate complex III₂V₂ respiratory supercomplex from *Mycobacterium smegmatis*. *Nat. Struct. Mol. Biol.* **25**, 1128-1136 (2018).
7. Birth, D., Kao, W.-C. & Hunte, C. Structural analysis of atovaquone-inhibited cytochrome bc_1 complex reveals the molecular basis of antimalarial drug action. *Nat. Commun.* **5**(2014).
8. Letts, J.A. *et al.* Structures of Respiratory Supercomplex I+III₂ Reveal Functional and Conformational Crosstalk. *Mol Cell* **75**, 1131-1146 e6 (2019).
9. Huang, L.-s., Cobessi, D., Tung, E.Y. & Berry, E.A. Binding of the Respiratory Chain Inhibitor Antimycin to the Mitochondrial bc_1 Complex: A New Crystal Structure Reveals an Altered Intramolecular Hydrogen-bonding Pattern. *J. Mol. Biol.* **351**, 573-597 (2005).

Additions to the SI:

Figure 1. Details of the structural model of the SC. i. The quinol was modelled in the Q_{o1a} site based on structural data of quinol and stigmatellin binding in canonical complex III (PDB IDs: 6Q9E⁸, 1PP9⁹, c.f. Extended Methods). The predicted binding site Q_{o1a} in the supercomplex (in blue) in comparison to ubiquinone (in red) and stigmatellin (in green) from the canonical Complex III.

5. Letts, J.A. *et al.* Structures of Respiratory Supercomplex I+III₂ Reveal Functional and Conformational Crosstalk. *Mol Cell* **75**, 1131-1146 e6 (2019).
9. Huang, L.-s., Cobessi, D., Tung, E.Y. & Berry, E.A. Binding of the Respiratory Chain Inhibitor Antimycin to the Mitochondrial bc_1 Complex: A New Crystal Structure Reveals an Altered Intramolecular Hydrogen-bonding Pattern. *J. Mol. Biol.* **351**, 573-597 (2005).

Question 2.1: Page 6:"In this regard, the Qo1a menaquinol is stabilised by His355 and Asp309, and water-mediated contacts with Tyr159, which in turn forms contacts with Asp302 of the characteristic PDWY motif 46,47 (Fig. 2b, 3a,b, Supplementary Fig. 9). Asp302 establishes an ion-pair with His110, the conformation of which modulates the binding affinity of the substrate (Supplementary Fig. 8c)."

- The described interactions should be shown in at least one figure, but unfortunately they are not in any of the mentioned ones. Moreover in Fig. 2b one hydroxyl group of menaquinol seems to interact with H355 and possibly with D309, which may accept the first proton, but according the figure, the second hydroxyl group is directed towards the F156, which rather questions the correct conformation of menaquinol at the Qo site.

Answer: We thank the Reviewer for pointing out that, while the plots in Figures S8 and S9e showed the distances between these residues from MD simulations, the previous version of Figure 2b did not clearly show these interactions. We have revised the figure, with a more representative figure of the menaquinol interacting with His355/Asp309 and Tyr159, and where the interaction between Tyr159 and Asp302 is also clearly visible. The revised figure also shows that D309 forms an ion-pair with R313 during the MD simulations that prevents direct proton transfer from the menaquinol to D309.

We have additionally noted and fixed a typo in "PDWY", which should be "PDFY" in this context.

Revised figure:

Figure 2. Quinone binding sites at the P-side of the membrane in the SC. a, Overview of the Q_o site, modelled based on PDB ID: 6hwh and 6adq, and supported by our current cryo-EM models. The sites were modelled with menaquinol at either the Q_{o1b} or at Q_{o2} site (panels c, d, respectively), based on resolved cryo-EM density, and also at the canonical Q_{o1a} site (panel b). Dashed lines highlight interactions that could be functionally relevant for catalysis. Superscripts denote residues from subunits other than QcrB, whilst an asterisk indicates residues, which do not interact directly with Q during the simulations.

Changes in the main text:

“In this regard, the Q_{o1a} menaquinol is stabilised by His355 and Asp309, and water-mediated contacts with Tyr159, which in turn forms contacts with Asp302 of the characteristic PDFY motif^{13,14} (Fig. 2b, 3a, Supplementary Fig. 9).”

47. Osyczka, A. et al. Role of the PEWY Glutamate in Hydroquinone–Quinone Oxidation–Reduction Catalysis in the Qo Site of Cytochrome bc_1 . *Biochemistry* **45**, 10492-10503 (2006).
48. Kao, W.-C. & Hunte, C. The Molecular Evolution of the Qo Motif. *Genome Biol. Evol.* **6**, 1894-1910 (2014).

Question 2.2: In Fig. 3b the water occupancies of QH₂ nad Qox are presented in graph, thus this figure is not a good reference for presenting binding mode of menaquinol. An unexplained abbreviation Qox was introduced in the drawing caption of Fig. 3b - is it quinone?

Answer: The text should not refer here to figure 3b and we have adjusted it accordingly. The Q_{ox} label does indeed refer to Q. We have changed the label to Q to be consistent with other figures.

Revised figure:

Figure 3. Functional hydration dynamics of the Complex III module of the SC. a, Menaquinol in the Q_{o1a} site as predicted by the MD simulations and the putative proton pathways towards Asp309 (path 1) and Asp302 (path 2) leading to the P-side bulk. b, Water occupancies from MD simulations for QH₂ and Q_{ox} (simulations S1/S7, Supplementary Table 1) along the pathways shown in panel a.

Question 3: Fig. 5 presents energy profiles for two steps of PCET reaction occurring at the Qo site. The barriers to these processes seem reasonable, but what is puzzling is the high instability of quinone compared to semiquinone and quinol. The text lacks an explanation for this result because if both quinol and semiquinone are more stable, then the formation of quinone is thermodynamically unfavorable. In the Extended methods/molecular models subsection Authors mentioned that MD simulations were performed for three possible position of menaquinol. The high energy of menaquinone obtained from QM/MM calculations indicates that binding manner of menaquinone might be even more interesting and worth testing.

Answer: We have now better clarified that the QM/MM simulations address the energetics of the local PCET reactions, when the menaquinol is already bound at the Q_{o1a} site. These free

energy profiles, do not account for the global energetics of diffusing into/out from this site. Based on the apparent K_m (120 μM , Ref. 2) and calculated binding energies (SI Fig. 8), the overall menaquinol binding from the membrane to the Q_o site(s) is exergonic ($\sim -4 \text{ kcal mol}^{-1}$), while the Q_{o1a} and Q_{o1b} sites are energetically comparable and exchangeable, which could provide further entropic stabilization for the binding step. Additionally, subsequent electron transfer reactions are overall exergonic (based on experimental redox potentials) and could drive the overall reaction forward.

It is also important to note that these enzymes operate under non-equilibrium high-flux conditions, where local endergonic steps are coupled to global exergonic reactions. This can be seen in our kinetic models, which also account for such differences in concentrations, and supports how the terminal oxygen reduction drives the complete charge transfer reaction. Moreover, the reduction level of the quinone pool and the proton motive force affect the effective rate of the quinol oxidation in Complex III. We added a schematic energy diagram in the SI that clarifies how these factors relate to the quinol oxidation and drive the reaction forward.

- Gong, H. *et al.* An electron transfer path connects subunits of a mycobacterial respiratory supercomplex. *Science* **362**, eaat8923 (2018).

Additions to the SI:

Figure S24. Schematic overview of charge transfer and quinone diffusion energetics. Representation of the energetics of the local PCET reaction at the Q_{1a} site, and the binding and release of the quinol/quinone from the membrane pool. Binding of quinol to the $QH_{2,o1b}$ and $QH_{2,o1a}$ sites from the membrane pool is exergonic by a few kcal mol^{-1} (based on $K_m \sim 120 \mu\text{M}$ (Ref.²) and calculated binding affinities (see also SI Fig. 8)). The two local binding sites could provide entropic stabilization into the binding process. The QM/MM calculations predict that the initial PCET is weakly exergonic, followed by an endergonic PCET towards heme b_L . The green curve accounts for the subsequent electron transfer and proton translocation processes, as well as the quinone unbinding, including also the cost for transferring the charges against the pmf (here 200 mV) based on experimental redox potentials¹². The subsequent electron transfer and proton translocation processes are exergonic, whilst the whole process is driven by the reduction of O_2 at the BNC ($E_m \sim 820 \text{ mV}$).

- Gong, H. *et al.* An electron transfer path connects subunits of a mycobacterial respiratory supercomplex. *Science* **362**, eaat8923 (2018).
- Kao, W.-C. *et al.* The obligate respiratory supercomplex from Actinobacteria. *Biochim. Biophys. Acta Bioenerg.* **1857**, 1705-1714 (2016).

Question 4.1: Page 8: "On the acceptor side of Complex III, we observe a subtle

motion of the Q_i quinone from the resolved binding position to a site, ca. 4 Å further from heme b_L, where the quinone forms a π-stacking interaction with Trp231 in the MD simulations (Fig. 3d, e). The heme b_L - heme b_H - Q_i network favours rapid electron transfer (10³-10⁴ s⁻¹, Fig. 6a), with our QM/MM simulations (Supplementary Fig. 14) further suggesting that Lys260 or Lys32 could function as local proton donors upon the quinone reduction (Supplementary Fig. 14, Movie 3), following uptake of the protons from the N-side via the nearby cardiolipin gate (see below, and cf. also Refs. 30,62)."

- Figure 3d is illegible, the mentioned lysines K260 and K32 seem to be quite distant from the quinone carbonyl groups as for proton donors - it is necessary to present hydrogen bonds here. Fig 14 in Supplementary shows transport of a proton from K260 via two water molecules, while K32 appears to be quite far from the second carbonyl group of the substrate. The figure shows the transport of a proton onto the Q₂⁻ ion, which is very unstable and such transport will certainly occur. Was it possible to obtain a semiquinone for this model, after all, in one Q_o cycle one electron is transferred to Q_i site. It is also necessary to present the energetics of the studied processes.

Answer: We thank the Reviewer for pointing out that Figure 3d required further improvement. We have thus removed some residues to reduce visual clutter and clearly marked putative proton transfer pathways.

QM/MM MD simulations of Q₂⁻ were used to assess the feasibility of this proton transfer and show that in this state proton transfer is strongly favoured, as expected. However, the Reviewer also raises a new interesting question about the behaviour of semiquinone at the Q_i site. From exploratory QM/MM simulations of the Q_i site, we can provide data that formation of the semiquinone species is not linked to a similar protonation transfer from the Lys, supporting the general notion that the proton uptake is redox coupled. Although we agree that these questions are interesting, they are outside the scope of the present work.

Changes in the main text:

The heme b_L - heme b_H - Q_i network favours rapid electron transfer (10³-10⁴ s⁻¹, Fig. 6a), with our QM/MM simulations (Supplementary Fig. 14) further suggesting that Lys260/Glu44 or Lys32 could function as local proton donors upon the quinone reduction (Supplementary Fig. 14, Movie 3a), following uptake of the protons from the N-side via the nearby cardiolipin gate (see below, and cf. also Refs.^{30,63}). In this regard, our simulations suggest that proton transfer may only occur after the Q_i species has been fully reduced (Supplementary Fig. 14, Movie 3a, b)

30. Postila, P.A. et al. Atomistic determinants of co-enzyme Q reduction at the Q_i-site of the cytochrome bc₁ complex. *Sci. Rep.* **6**, 33607 (2016).

63. Lange, C., Nett, J.H., Trumpower, B.L. & Hunte, C. Specific roles of protein-phospholipid interactions in the yeast cytochrome bc₁ complex structure. *EMBO J.* **20**, 6591-600 (2001).

Revised figures:

Figure 3. Functional hydration dynamics of the Complex III module of the SC. *a*, Menaquinol in the Q_{o1a} site as predicted by the MD simulations and the putative proton pathways towards Asp309 (path 1) and Asp302 (path 2) leading to the P-side bulk. *b*, Water occupancies from MD simulations for QH_2 and Q_{ox} (simulations S1/S7, Supplementary Table 1) along the pathways shown in panel *a*. *c*, MD-based water clustering and resolved cryo-EM water molecules predict similar proton pathways from the Q_{o1a} site to the P-side bulk. *d*, The quinone at the Q_i site forms a π -stacking interaction with Trp231, and two possible proton pathways from the N-side bulk via Lys260 and Lys32. The experimentally resolved starting position of the quinone is shown in light green.

Addition to the SI:

Movie S3. Protonation dynamics upon reduction of Q_i based on QM/MM MD simulations. With *a*) Q^{2-} and *b*) Q^{1-} species.

Question 4.2: It is difficult to verify parameters used in MD simulations. Methods used are presented, but there is a lack of information on the structure of the models used to parameterize the redox cofactors. The figures presenting such models should be included in the Supplementary. This is important since in the case of FeS clusters, the protein surrounding significantly affects the cofactor parameters, its geometry and partial charges, while in the case of hemes, taking into account propionate groups without taking into account the strong electrostatic interactions they form with protein also distorts their bonding and nonbonding parameters, and thus affects the MD results. Correct parameterization of redox cofactors in the cases of the considered reactions in which they influence the substrate binding and what is more important directly participate in catalysis is crucial to obtain reliable results of MD simulation.

Answer: We understand the Reviewers concerns but would like to point out that the extended methods section “Derivation of force field parameters” already clearly states that these models are shown in the SI Figure 21:

“See Supplementary Figure S21 for model systems used for parametrisation, and interaction validation as well as Supplementary tables S7-S11 and Refs¹⁰ for force field parameters.”

Force field parameters are generally developed to be transferable, thus implying that large parts of the protein surroundings cannot be included in the parametrisation procedure (as this would make the parameters protein isoform specific). As explained in the Extended Methods section, we have in this regard made exceptions to this rule for the unique binuclear centre, where the protein surroundings were considered. The quinones have not been parametrised for each binding site specifically, as this would bias the simulations. The *b* and *c* type of hemes similarly follow the CHARMM parameters.

To further support the correctness of the force field parameters, we have included a comparison of geometries obtained by MM and QM/MM minimisation (SI Figure S23 and Tables S12-14) that show good agreement. We also wish to point out that the QM/MM exploration of the catalysis does not rely on any force field parameters, as the interactions of the reacting Q species, the FeS, and heme are calculated on the fly at quantum mechanical level.

Addition to the Extended Methods:

See Supplementary Figure S21 for model systems used for parametrisation, and interaction validation as well as Supplementary tables S7-S14 and Refs¹⁰ for force field parameters.

10. Johansson, M.P., Kaila, V.R.I. & Laakkonen, L. Charge parameterization of the metal centers in cytochrome *c* oxidase. *J. Comput. Chem.* **29**, 753-767 (2008).

Addition to the SI:

Figure 23. Force field validation. Comparison of differences in bond lengths and angles between MM and QM minimised structures of a, b) quinone, c, d) FeS centre, and e, f) heme *b*. See also Tables S12-S14. The MM and QM/MM optimisations were initiated from independent snapshots followed by a geometry optimization. The MM system was minimised for 10000 steps, while the QM/MM system was minimised until energy convergence threshold of $0.006 \text{ kcal mol}^{-1}$, with the MM region restrained.

Table S12. Validation of the menaquinone geometry.

Table S13. Validation of the FeS geometry.

Table S14. Validation of heme *b* geometry.

Question 5: The article lacks discussion on the protonation state of residues directly interacting with quinol/quinone and their impact on the binding of the reactant and the catalytic abilities of the enzyme. Were the protonation states of H355, D302, D309, H110 verified in any way? What are the pKa values of these residues? A similar question arises about protein residues interacting with the quinone at the Qi site.

Answer: As stated in the Method section “Molecular dynamics simulations”, initial protonation states were assigned using “Poisson-Boltzmann electrostatic calculations with Monte Carlo sampling”, and the protonation states of these residues were reported in Table S1. We have further provided a new Table with explicitly calculated pKa values (SI Table S15).

Notably, His110 seems to be able to adopt both protonated and deprotonated states based on its conformation, so we chose to probe both states to explore how these affect the quinone binding/catalysis. Lys260 is predicted to be protonated based on the modelled cryo-EM conformation, where it has no available interaction partner. Dynamics, on the other hand, show that it can readily interact with Glu44, likely favouring a protonated state, which we ended up modelling.

Addition to the Extended Methods:

Initial protonation states were obtained from using Poisson-Boltzmann electrostatic calculations with Monte Carlo sampling^{13,14}, based on models with the cofactors treated in their oxidised state (see Supplementary Table 3 and Table 15, for a list of residues with non-standard protonation states and their estimated pK_a values, respectively).

13. Kieseritzky, G. & Knapp, E.W. Optimizing pK_a computation in proteins with pH adapted conformations. *Proteins* **71**, 1335-48 (2008).
14. Kieseritzky, G. & Knapp, E.W. Improved pK_a prediction: combining empirical and semimicroscopic methods. *J. Comput. Chem.* **29**, 2575-81 (2008).

Addition to the SI:

Table S15. Calculated pK_a values of selected residues in the Q_{o1a} and Q_i sites. The calculations were performed for Complex III (PDB ID: 6ADQ)² with bound quinone before (6adq) and after (6adq opt) geometry optimisation of sidechains. The table reports values for each protomer.

Residue	6adq	6adq opt
Lys32	9.5/9.4	2.7/2.8
Glu44	9.7/9.7	1.5/5
His110	>14/>14	<0/<0
Lys260	1.7/1.8	3.5/3.4
Asp302	0.0/0.0	<0/<0
Asp309	3.6/3.9	2.3/2.0

2. Gong, H. *et al.* An electron transfer path connects subunits of a mycobacterial respiratory supercomplex. *Science* **362**, eaat8923 (2018).

Reviewer #1 (Remarks to the Author):

The authors have addressed all my comments and the manuscript should be recommended for publication.

Reviewer #2 (Remarks to the Author):

The manuscript has been revised and supplemented. As far as possible, based on the collected results, the Authors addressed the questions and doubts I raised. Figure 2b has been revised, but introduction of the key distance values would be useful. A similar revision would be useful for Figure 2c and d with the Qo1b and Qo2 binding sites, because in the current version these Figures are not very informative. According to the suggestion, the manuscript was also supplemented with the conclusion section, in which the Authors summarized the obtained results. I however uphold my opinion that the manuscript does not bring new significant advances with respect to mechanistic discussion on complexes III and IV. It rather confirms data and notions already existing in the literature. I neither see structural data that would add new significant information to what has already been described in previously published cry-EM structures of this complex.

Reviewers' Comments:

Comment: The manuscript has been revised and supplemented. As far as possible, based on the collected results, the Authors addressed the questions and doubts I raised. Figure 2b has been revised, but introduction of the key distance values would be useful. A similar revision would be useful for Figure 2c and d with the Qo1b and Qo2 binding sites, because in the current version these Figures are not very informative.

According to the suggestion, the manuscript was also supplemented with the conclusion section, in which the Authors summarized the obtained results. I however uphold my opinion that the manuscript does not bring new significant advances with respect to mechanistic discussion on complexes III and IV. It rather confirms data and notions already existing in the literature. I neither see structural data that would add new significant information to what has already been described in previously published cry-EM structures of this complex.

Answer: We have updated the figure caption to clearly refer the reader to SI Figure 8, which not only shows example values, but also distances between all residues and the quinone during the full trajectories.